# Transport-limited fluvial erosion – simple formulation and efficient numerical treatment

Stefan Hergarten[1]

[1]Institut für Geo- und Umweltnaturwissenschaften, Albertstr. 23B, 79104 Freiburg, Germany

**Correspondence:** Stefan Hergarten
(stefan.hergarten@geologie.uni-freiburg.de)

**Abstract.** Most of the recent studies modeling fluvial erosion in the context of tectonic geomorphology focus on the detachment-limited regime. One reason for this simplification is the simple relationship of the constitutive law used here – often called stream-power law – to empirical results on longitudinal river profiles. Another, not less important reason lies in the numerical effort that is much higher for transport-limited models than for detachment-limited models. This study proposes a formulation of transport-limited erosion where the relationship to empirical results on river profiles is almost as simple as it is for the stream-power law. As a central point, a direct solver for the fully implicit scheme is presented. This solver requires no iteration for the linear version of the model, allows for arbitrarily large time increments, and is almost as efficient as the established implicit solver for detachment-limited erosion. The numerical scheme can also be applied to linear hybrid models that cover the range between the two end members of detachment-limited and transport-limited erosion.

## 1 Introduction

Rivers play a major if not dominant part in large-scale landform evolution. If horizontal displacement of the crust is not taken into account, models describing the evolution of a topography $H(x_1, x_2, t)$ are typically written in the form

$$\frac{\partial H}{\partial t} = U - E, \tag{1}$$

where $U$ and $E$ are the rates of uplift of crustal material relative to a given datum and of erosion, respectively.

Two end members – detachment-limited and transport-limited erosion – are widely considered in the context of fluvial landform evolution. The term "detachment-limited erosion" was presumably coined by Howard (1994). The idea behind this concept is that all particles entrained by the river are immediately removed from the system. The erosion rate $E$ can be considered as a function of local properties at each point. In the simplest approach, these are catchment size and channel slope (slope in direction of steepest descent), while all other influences are subsumed in a lumped parameter often called erodibility.

In all scenarios other than the detachment-limited case, a sediment balance must be considered. If no material is directly removed, the erosion rate is

$$E = \text{div}\boldsymbol{q}, \tag{2}$$

where $\boldsymbol{q}$ is the sediment flux per unit width (volume per time and cross section length) and div the 2-D divergence operator. It is usually assumed that $\boldsymbol{q}$ follows the direction of the channel slope, so only its absolute value $q$ varies between different models.

The concept of transport-limited erosion assumes that the rate of bed erosion is limited by the ability of the flow to transport the eroded material, rather than by the availability of potentially mobile sediment. The implementation of this concept in fluvial landform evolution models presumably dates back to Willgoose et al. (1991b). Transport-limited models directly define the sediment flux per unit width $q$ instead of the erosion rate $E$ at each point as a function of local properties such as catchment size and channel slope.

Mathematically, both concepts differ fundamentally. Equation (1) only involves derivatives of first order with regard to time and with regard to the spatial coordinates (arising from the channel slope) in the detachment-limited scenario. So it is a hyperbolic differential equation of the advection type. Propagation of information in one direction only – upstream here – is a characteristic property of this type. Anything that happens at a given point and a given time only affects the region upstream of this point in the future. In contrast, Eq. (1) contains spatial derivatives of second-order in the transport-limited regime since

$q$ inside the divergence operator depends on the channel slope. Equation (1) combined with Eq. (2) is a parabolic differential equation of the diffusion type then, where information propagates in both upstream and downstream direction.

Several comprehensive numerical models of fluvial landform evolution have been developed since the 1990s. All models reviewed by Coulthard (2001), Willgoose (2005), and van der Beek (2013) involve a sediment balance. In recent years, however, using the detachment-limited model has become a popular choice, although the idea that all particles are immediately excavated is limited has been questioned (e.g., Turowski, 2012). All types of bedload transport are obviously not captured by this concept.

Nevertheless, even some recent studies using models that are able to simulate sediment transport focus on the detachment-limited case (e.g., Duvall and Tucker, 2015; Theodoratos et al., 2018; Eizenhöfer et al., 2019).

At least three aspects make the detachment-limited approach appealing. First, the relationship to empirical studies of longitudinal channel profiles is particularly simple here. Hack (1957) observed a power-law relationship between channel slope $S$ and upstream catchment size $A$ in several rivers. This relationship is nowaday often called Flint's law (Flint, 1974) and written

in the form

$$S = k_s A^{-\theta}, \tag{3}$$

where $\theta$ is the concavity index and $k_s$ the steepness index. Assuming that Eq. (3) is the fingerprint of a spatially constant erosion rate under uniform conditions, it can be assumed that

$$E = f(k_s) = f(A^\theta S), \tag{4}$$

where $f$ is an arbitrary function. A power-law function

$$f(k_s) = K k_s^n = K(A^\theta S)^n, \tag{5}$$

where the parameter $K$ is denoted erodibility, is a common choice in this context. The fluvial erosion rate is often written in the form

$$E = K A^m S^n \tag{6}$$

with $m = \theta n$. Equation (6) is often called stream-power law and its combination with Eq. 1 stream-power incision model since it can be interpreted in terms of energy dissipation of the water per channel bed area if an empirical relationship between channel width and catchment size is used (e.g., Whipple and Tucker, 1999).

The concavity index $\theta = \frac{m}{n}$ appears to be well constrained, so most modeling studies either use the value $\theta = 0.5$ originally found by Hack (1957) or a slightly lower reference value $\theta = 0.45$ (e.g., Whipple et al., 2013; Lague, 2014). In turn, the value of the exponent $n$ is less well constrained since it cannot be determined from the shape of equilibrium profiles under uniform conditions. The model is linear with regard to $H$ (if the flow pattern is given) for $n = 1$, which simplifies both theoretical considerations and the numerical implementation. Thus, the remaining uncertainty in the effective value of $n$ often serves as a reason for choosing $n = 1$, although, e.g., the results compiled by Lague (2014) rather suggest $n > 1$. If $\theta$ is well constrained and $n = 1$ is accepted as a convenient choice, the erodibility $K$ remains as the only parameter. It is a lumped parameter subsuming all influences on erosion other than channel slope and catchment size. So it is not only a property of the rock, but also depends on climate in a nontrivial way (e.g., Ferrier et al., 2013; Harel et al., 2016). However, it just defines how steep rivers will become at a given uplift rate, so reasonable values can be found, e.g., by analyzing river profiles at situations where estimates of the uplift rate are available.

Constitutive laws based on power-law relations, however, have not been employed only in detachment-limited models. Even the earliest numerical model of transport-limited erosion (Willgoose et al., 1991b) used a power law for the sediment flux density based on physical relations for the shear stress at the bed. The empirical results on real rivers represented by Eq. (3) were also used to constrain the parameter values before the detachment-limited concept became popular (Willgoose et al., 1991a). However, the transport-limited approach never reached the simplicity of the detachment-limited approach with regard to the small number of parameters and their quite direct relation to the properties of real river profiles.

The simplicity of the differential equation itself serves as a second argument in favor of the detachment-limited approach. In the linear case ($n = 1$), Eq. (1) combined with Eq. (6) can be solved analytically for any given uplift pattern and history. The term $KA^\theta$ defines the velocity of advection then, so disturbances propagate in upstream direction at a this velocity. The treatment can be simplified by the $\chi$ transform introduced by Perron and Royden (2013). It transforms the upstream coordinate $x$ to a new coordinate

$$\chi = \int \left( \frac{A(x)}{A_0} \right)^{-\theta} dx, \tag{7}$$

where $A_0$ is an arbitrary reference catchment size and the integration starts from an arbitrary reference point. This transformation eliminates the inherent curvature of river profiles arising from the decrease of catchment size in upstream direction, so equilibrium profiles under spatially uniform conditions turn into straight lines. The solutions of this equation and their potential for unraveling the uplift and erosion history were investigated by Royden and Perron (2013), and a formal inversion procedure for the linear case ($n = 1$) was presented by Goren et al. (2014). So the detachment-limited model can be reconciled with real river profiles not only under steady-state conditions, but also in the context of temporal changes.

As a third point, detachment-limited erosion can be implemented in numerical models more efficiently than transport-limited erosion. Here, even a fully implicit scheme that allows for arbitrary time increments with linear time complexity, also known

as $O(N)$, is available. This means that the computing effort increases only linearly with the total number of nodes $N$. The scheme was introduced in the context of fluvial erosion by Hergarten and Neugebauer (2001), described in detail for $n = 1$ and $n = 2$ by Hergarten (2002), and made popular by Braun and Willett (2013).

So far there is no comparable implementation for transport-limited erosion. As mentioned above, transport-limited erosion corresponds to a diffusion-type equation. The challenge is that the diffusivity depends on the catchment size and thus varies over several orders of magnitude. Multigrid methods (e.g., Hackbusch, 1985) are still the only schemes for the diffusion equation in more than one dimension with linear time complexity. However, convergence breaks down if the diffusivity varies by some orders of magnitude, so multigrid methods have not been applied in the context of fluvial erosion. So far none of the existing landform evolution model treat the transport-limited case with a fully implicit scheme that allows for arbitrarily large time increments.

The advantage of the detachment-limited model concerning the numerical complexity persists if explicit schemes are used here, too. The main reason for using explicit schemes for detachment-limited erosion is the artificial smoothing of knickpoints by the implicit discretization, while explicit schemes that preserve the shape of knickpoints better are available. A comparison was given by Campforts et al. (2017). As already pointed out by Howard (1994), explicit schemes for the transport-limited case typically require 3 to 4 orders of magnitude shorter time steps than for the detachment-limited case.

Howard (1994) already developed an approximation that makes the explicit scheme for the transport term numerically more stable. Kooi and Beaumont (1994) proposed an approach that increases stability and also allows for a physical interpretation, often called undercapacity model or – in a more general context – linear decline model (Whipple and Tucker, 2002). It defines an equilibrium flux per unit width $q_e$ from local properties (channel slope, catchment size, . . . ) and assumes that the erosion rate is

$$E = \frac{q_e - q}{l}. \tag{8}$$

The parameter $l$ defines a length scale and can be seen as inertia of sediment detachment and deposition against changes in fluvial conditions.

An alternative physical interpretation of the linear decline model was developed by Davy and Lague (2009). The detachment-limited model (Eq. 6) was extended by a sediment deposition term proportional to the actual sediment flux. As a main point, Davy and Lague (2009) found an expression for the rate of deposition that keeps equilibrium river profiles consistent with Eq. (3), which is not the case for the original undercapacity model (Whipple and Tucker, 2002).

Yuan et al. (2019) implemented an implicit numerical scheme for this model based on a Gauss-Seidel iteration in the upstream direction. The rate of convergence was found to be independent of the size of the grid, so the scheme is indeed of linear time complexity. However, the convergence slows down strongly for faster deposition, i.e., when approaching the transport-limited end member. The scheme of Yuan et al. (2019) is therefore presumably the most efficient implementation of sediment transport in large-scale fluvial erosion models, but achieves its full power only if we do not come too close to the transport-limited end member.

In the following section, a formulation of transport-limited erosion is proposed that can be directly reconciled with the concept of the erodibility. Then, Sect. 3 presents a fully implicit, direct scheme for solving the equation numerically. After presenting a numerical example in Sect. 4, Sect. 6 provides a discussion several versions of the linear decline model and an extension of the numerical scheme for this class of models.

## 2 Simple formulation of transport-limited erosion

Let us start from the interpretation of Hack's empirical relation (Eq. 3) as the fingerprint of uniform erosion under spatially constant conditions, regardless of the mechanism of erosion. This implies that the erosion rate is a function of the steepness index (Eq. 4). Then the sediment flux $Q$ (volume per time, not per unit width) through any cross section of a river is

$$Q = \int E \, dA, \tag{9}$$

where the integral extends over the upstream catchment. For uniform erosion, the integral reduces to the product of the erosion rate and the catchment size,

$$Q = AE = A f(A^\theta S). \tag{10}$$

If a power-law function (Eq. 6) is used in analogy to the detachment-limited model, the sediment flux becomes

$$Q = K A^{m+1} S^n. \tag{11}$$

In contrast to the more common formalism based on the flux per unit width $q$ (Eq. 2), these relations use the total sediment flux $Q$ (volume per time) passing the entire cross section of a channel segment. This total flux cannot be inserted formally into the divergence operator in Eq. (2) to form a continuous differential equation. Practically, however, this is not a problem for a discrete channel network. If any pixel of the considered topography has a unique drainage direction towards a single neighbor and sediment transport follows flow direction, the respective discrete version of the divergence operator at the node $i$ is

$$\mathrm{div}\boldsymbol{q}_i = \frac{Q_i - \sum_j Q_j}{s_i}, \tag{12}$$

where $Q_i$ is the flux from the node $i$ to its flow target. The sum extends over all neighbors that deliver their discharge und thus their sediment flux to the node $i$, called donors in the following. Finally, $s_i$ is the area of the considered node, i.e., the pixel size for a regular mesh or the area of the respective cell in a general finite-volume discretization. As the model describes the total sediment flux and not flux per unit width, an integration over the edges of the cell is not necessary.

Inserting Eqs. (2) and (12) into Eq. (1) then yields the simplest form of a transport-limited fluvial erosion model,

$$s_i \frac{\partial H_i}{\partial t} = s_i U_i - Q_i + \sum_j Q_j, \tag{13}$$

where $Q_i$ is defined by Eq. (10) or Eq. (11).

As mentioned above, using power-law functions for sediment transport is not new. In combination with empirical relations for the channel width, physically-based relations for the sediment flux density (e.g., Willgoose et al., 1991b) support the hypothesis of a power-law dependence of $Q$ on $A$ and $S$ (Eq. 11). However, the relations where never written in such a simple form as in Eq. (11) with parameters that are related so closely to the concepts of concavity index and steepness index (Eq. 3). Equation (11) was discussed in the literature (e.g., Whipple and Tucker, 2002) in the context of equilibrium river profiles, but apparently never used directly for defining a transport-limited erosion model. In view of Hack's findings this is, however, as straightforward as describing detachment-limited erosion by Eq. (4) or Eq. (6). Even the physical unit of the erodibility $K$ is the same in both models, and the same values of $K$ yield the same erosion rate at the same topography for spatially uniform erosion.

The two models are, however, not equivalent for non-uniform erosion. According to Eqs. (4) and (5), the steepness index $k_s$ directly reflects the erosion rate at the considered point in the form

$$K k_s^n = E \tag{14}$$

for the detachment-limited model. In turn, Eq. (11) of the transport-limited model can be combined with Eq. (9) to

$$K k_s^n = \frac{Q}{A} = \frac{1}{A} \int E \, dA. \tag{15}$$

This relation is basically the same as Eq. (14) except that the right-hand side is the mean erosion rate of the upstream catchment instead of the local erosion rate at the considered point. So channel steepness directly reflects the local erosion rate in the detachment-limited model, but the mean erosion rate of the catchment for the transport-limited model.

The same holds for the interpretation of the erodibility $K$. In the detachment-limited model, it describes how much material is eroded at the considered location for a given steepness index. In turn, it describes how much material is eroded on average in the upstream catchment in the transport-limited model. From a process-oriented point of view, $K$ would rather be considered a transport coefficient than an erodibility here. However, this is just a matter of terminology.

## 3   A fully implicit numerical algorithm for transport-limited erosion

The model proposed in the previous section can be treated with an efficient, fully implicit numerical scheme in the linear case ($n = 1$). The reason why this is possible in contrast to the 2-D diffusion equation lies in the tree structure of the flow and sediment transport pattern.

The fully implicit discretization of Eq. (13) reads

$$s_i \frac{H_i(t) - H_i(t_0)}{\delta t} = s_i U_i - Q_i(t) + \sum_j Q_j(t), \tag{16}$$

where the time step extends from $t_0$ to $t$ and $\delta t = t - t_0$. The solution at $t_0$ is known, and the solution at $t$ is computed.

Let the node $b$ be the flow target of the node $i$, so $H_b$ serves as a base level for the node $i$. As the entire problem is linear, the height $H_i$ responds linearly to base level changes. Figure 1 illustrates this behavior in a simple numerical example of a river

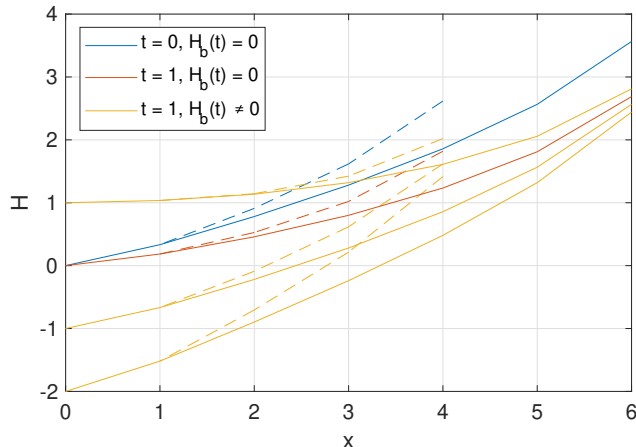

**Figure 1.** River profiles obtained from one implicit time step, where all parameters ($K$, $\delta t$, grid spacing) are set to unity. The blue line describes a steady state with $U = 1$, and it is assumed that $U = 0$ for $t > 0$.

(solid lines) with one tributary (dashed lines). The initial state ($t = 0$, blue) was a steady state under constant uplift. The red curves ($t = 1$) show the result of an implicit time step without uplift and with the same base level ($H_b = 0$) as for $t = 0$, while the orange curves correspond to different base levels $H_b$. The four red and orange curves of each river are equidistant at each point $x$ for equal increments in $H_b$, so the change in height $H_i(t)$ due to changes in base level is proportional to the change in base level $H_b(t)$.

Due to the linearity, the sediment flux $Q_i$ to the node $b$ also responds linearly to base level changes and can therefore be written in the form

$$Q_i(t) = Q_i^0 + Q_i'\left(H_b(t) - H_b(t_0)\right). \tag{17}$$

Here, $Q_i^0$ is the flux that occurs if the base level $H_b$ remains constant ($H_b(t) = H_b(t_0)$), and $Q_i'$ is the derivative of $Q_i(t)$ with regard to base level changes. Inserting Eq. (17) for the donors into Eq. (16) yields

$$s_i \frac{H_i(t) - H_i(t_0)}{\delta t} = s_i U_i - Q_i(t) + \sum_j Q_j^0 + \sum_j Q_j'\left(H_i(t) - H_i(t_0)\right) \tag{18}$$

and thus

$$Q_i(t) + \frac{\alpha_i}{\delta t}\left(H_i(t) - H_i(t_0)\right) = \beta_i \tag{19}$$

with the terms

$$\alpha_i = s_i - \delta t \sum_j Q_j' \quad \text{and} \quad \beta_i = s_i U_i + \sum_j Q_j^0 \tag{20}$$

introduced in order to keep the equations short. Similarly to the detachment-limited model, nodes without any donors act as boundaries within the domain. These nodes do not require any specific treatment except that the respective sums in Eqs. (18) and (20) are empty.

The channel slope at the node $i$ is

$$S_i(t) = \frac{H_i(t) - H_b(t)}{d_i},\tag{21}$$

where $d_i$ is the distance between the nodes $i$ and $b$. So the sediment flux is

$$Q_i(t) = KA_i^{m+1}\frac{H_i(t) - H_b(t)}{d_i}\tag{22}$$

according to Eq. (11) for $n = 1$. This leads to

$$H_i(t) = H_b(t) + \frac{d_i}{KA_i^{m+1}}Q_i(t).\tag{23}$$

Inserting this relation into Eq. (19) yields

$$Q_i(t) + \frac{\alpha_i}{\delta t}\left(\frac{d_i}{KA_i^{m+1}}Q_i(t) + H_b(t) - H_i(t_0)\right) = \beta_i,\tag{24}$$

which can be rearranged in the form

$$Q_i(t) = \frac{\alpha_i\left(H_i(t_0) - H_b(t)\right) + \beta_i\delta t}{\alpha_i\frac{d_i}{KA_i^{m+1}} + \delta t}.\tag{25}$$

Comparing this expression with Eq. (17) yields

$$Q_i^0 = \frac{\alpha_i\left(H_i(t_0) - H_b(t_0)\right) + \beta_i\delta t}{\alpha_i\frac{d_i}{KA_i^{m+1}} + \delta t}\tag{26}$$

and

$$Q_i' = -\frac{\alpha_i}{\alpha_i\frac{d_i}{KA_i^{m+1}} + \delta t}.\tag{27}$$

Equations (26) and (27) allow for the computation of $Q_i^0$ and $Q_i'$ from the respective values of the donors (because $\alpha_i$ and $\beta_i$ depend on these) and from known elevation values at the time $t_0$. All values $Q_i^0$ and $Q_i'$ can thus be computed successively in downstream direction. As the required order of the nodes is the same as for computing the catchment sizes $A_i$, it is most efficient to calculate $Q_i^0$ and $Q_i'$ in the same sweep over the nodes where the catchment sizes are computed.

Once the values $Q_i^0$ and $Q_i'$ have been computed for all nodes, the sediment flux $Q_i(t)$ can be computed using Eq. (17). This sediment flux is then used for computing the elevation $H_i(t)$ from Eq. (23). As these steps require the elevation of the flow target $H_b(t)$, they have to be performed successively in upstream order. This order is the same as used in the implicit scheme for detachment-limited erosion.

So the numerical scheme consists of three sweeps over the grid:

**Sweep 1:** Compute the flow directions $b$ of all nodes. The nodes can be processed in any order.

**Table 1.** Time complexity of the scheme for transport-limited erosion compared to the implicit scheme for detachment-limited erosion. CPU time was normalized to the total effort of one time step for detachment-limited erosion.

|  | Detachment limited | | Transport limited | |
|---|---|---|---|---|
|  | properties | CPU time (%) | properties | CPU time (%) |
| sweep 1 | $b$ | 38 | $b$ | 38 |
| sweep 2 | $A$ | 49 | $A, Q^0, Q'$ | 54 |
| sweep 3 | $H$ | 13 | $Q, H$ | 20 |
| total |  | 100 |  | 112 |

**Sweep 2:** Compute the catchment size $A$ and the properties $Q^0$ (Eq. 26) and $Q'$ (Eq. 27) of all nodes. The nodes have to be processed in downstream order. This is implemented most conveniently in a recursive scheme with a function that computes the three above properties for each node. Before computing these values, the function checks which of the donors have already been treated and invokes itself for those donors that have not been considered before.

**Sweep 3:** Compute $Q(t)$ according to Eq. (17) and $H(t)$ from Eq. (23) for all nodes. The nodes must be processed in upstream order, which is also performed conveniently by a recursive implementation. The principle is the same as in sweep 2 except that the flow target has to be considered instead of the donors.

The scheme is a direct scheme without any iterative component. The derivatives $Q'$ are always negative (lower base level leads to a higher sediment flux), so that the properties $\alpha$ and thus the denominator in Eqs. (26) and (27) are always positive. So the scheme is unconditionally stable, and its time complexity is linear ($O(N)$) under all conditions.

The workflow with the three sweeps is basically the same as in the implicit scheme for detachment-limited erosion. The structure is the same without any extra loops, conditions or functions to be invoked. Additional effort only arises from floating-point operations. Table 1 provides an estimate of the time complexity compared to detachment-limited erosion. All results were obtained using the landform evolution model OpenLEM that was used in some previous studies (e.g., Robl et al., 2017; Wulf et al., 2019; Hergarten, 2020a), but has not been published explicitly. A regular $5000 \times 5000$ grid was used, and the results of several runs involving 100 to 1000 time steps were checked for consistency. The CPU time was normalized to the total effort of one time step for detachment-limited erosion. The difference in time complexity between both models is marginal.

With regard to memory complexity, the scheme presented here requires two additional variables per node, $Q^0$ and $Q'$. When performing the third sweep, one of them can be recycled for storing the original surface height $H(t_0)$ that is needed later when Eq. (17) is applied to the donors. The remaining variable can be used for storing the actual sediment flux $Q(t)$ in case it is needed later.

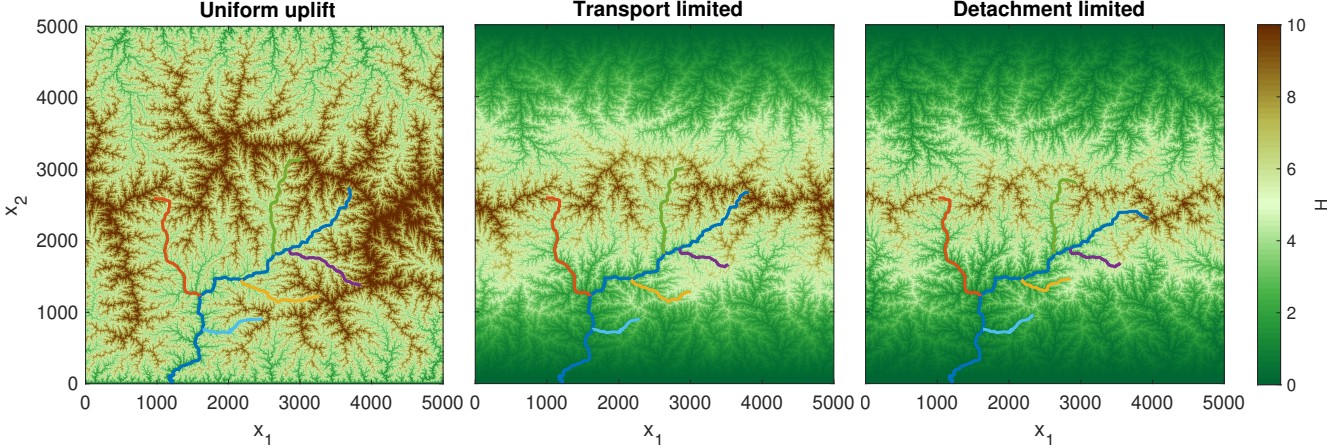

**Figure 2.** Equilibrium topographies for uniform uplift (left) and for a tent-shaped uplift pattern (middle and right). The color-coded rivers are the largest stream and its 5 largest tributaries in the topography for uniform uplift. They are referred to in Fig. 4.

## 4 A numerical example

The transport-limited model proposed in Sect. 2 is equivalent to the detachment-limited model only for uniform erosion. Transient states are typically characterized by spatially variable erosion, so the two end members cannot cannot yield the same transient behavior. This result is, however, already clear from more general arguments since both end members are described by differential equations of different types (parabolic vs. hyperbolic) as discussed in Sect. 1.

This section presents numerical example showing that non-uniform conditions result in strong differences between the two models even in a steady state. The example uses a square domain of $5000 \times 5000$ nodes. The northern and southern boundaries are kept at $H = 0$, while the two other boundaries are periodic. All horizontal lengths and areas are measured in terms of pixels. An exponent $m = 0.5$ was assumed, so that equilibrium rivers have a concavity index of $\theta = 0.5$ for the linear model ($n = 1$). The erodibility was set to $K = 1$.

An equilibrium topography obtained for uniform uplift $U = 1$ was used as a reference. This topography (Fig. 2, left) was generated by starting from a flat initial topography with a small random disturbance. As the transport-limited and the detachment-limited models are equivalent for uniform erosion, this topography is an equilibrium topography for both models.

As a simple non-uniform uplift pattern, tent-shaped uplift is considered. The maximum uplift rate $U = 1$ is achieved here in the middle between the northern and southern boundary ($x_2 = 2500$) and decreases linearly to zero towards the boundaries. In order to get similar flow patterns (Fig. 2), the equilibrium topography corresponding to constant uplift was used as an initial condition.

All equilibrium topographies were computed by starting with small increments $\delta t$ that are increased through time when the number of changes in flow direction per time step is sufficiently small. This procedure is useful for generating steady-state topographies with similar large-scale flow patterns at a reasonable number of time steps. At large $\delta t$, smaller random values of

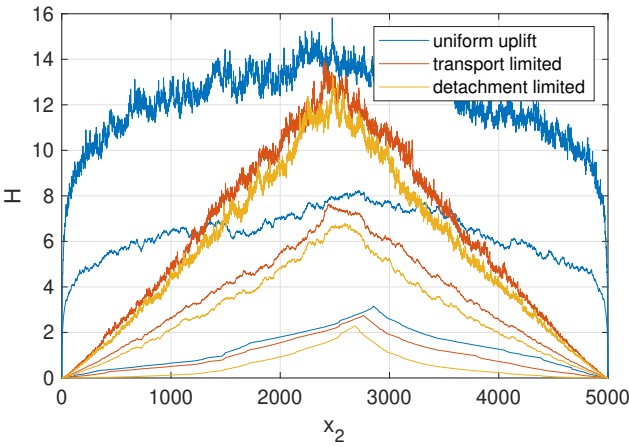

**Figure 3.** Swath profiles through the topographies shown in Fig. 2. The three lines of each color describe maximum, mean, and minimum elevation in east-west direction, i.e., over all values of $x_1$.

$\delta t$ were used in each second step in order to avoid periodic oscillations between topographies with different flow patterns that prevent the topography from reaching a steady state.

The tent-shaped uplift pattern causes an overall increase in uplift in upstream direction at least for large rivers. This increase results in an upstream increase of steepness. As the steepness reflects the mean erosion rate of the upstream catchment (Eq. 15) instead of the local erosion rate for transport-limited erosion, it varies more gently with the uplift rate here than for detachment-limited erosion.

Figure 3 shows swath profiles through the three topographies. The maximum surface height (uppermost curve of the respec-
tive color) is dominated by the steep slopes at small catchment sizes. Since these depend on the local uplift rate in equilibrium, the maximum elevation roughly follows the tent-shaped uplift pattern with minor differences between transport-limited and detachment-limited erosion. The absolute difference between the two models is similar for maximum, mean, and minimum elevation, so it can be attributed to the different heights of large valleys, while local relief is similar.

The profiles of the large rivers marked in Fig. 2 are shown in Fig. 4. For a clearer representation, the longitudinal coordinate
was $\chi$ transformed according to Eq. (7) with $A_0 = 1$. With the value $K = 1$ used here, equilibrium profiles follow a straight line $H = \chi$ at a uniform uplift rate $U = 1$. In turn, $\chi$-transformed equilibrium profiles are concave if the uplift rate increases in upstream direction. This concavity is weaker for the transport-limited model than for the detachment-limited model as the local slope reflects the mean erosion rate of the upstream catchment, while it reflects the local erosion rate for detachment-limited erosion. In the upper part of the catchment, however, both turn into parallel straight lines. In the lower part of the catchment,
the river profiles of the transport-limited model are steeper than those of the detachment-limited model because the river also has to carry away the material from the upper part with high erosion rates.

While the $\chi$-transformed river profiles of the transport-limited model are more straight than for detachment-limited erosion, local collinearity of tributaries is lost. For detachment-limited erosion, profiles of tributaries start with the same slope as the

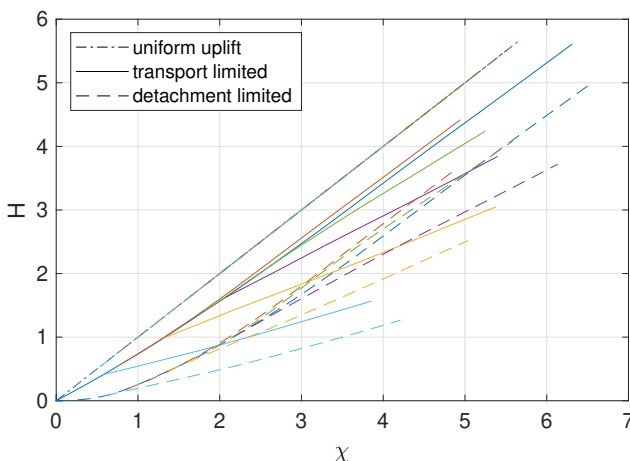

**Figure 4.** Longitudinal profiles of the rivers marked in Fig. 2 plotted in $\chi$ representation.

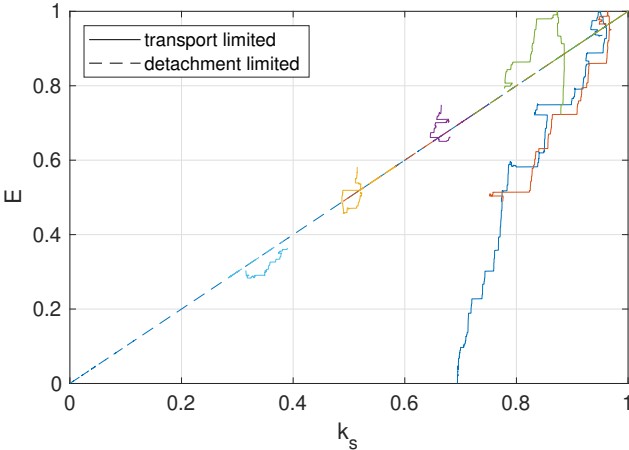

**Figure 5.** Erosion rate $E\ (=U)$ vs. steepness index $k_s$ for the rivers marked in Fig. 2.

trunk stream and deviate more and more with increasing distance. In contrast, tributaries and the trunk stream may contribute
different amounts of sediment per catchment size due to different mean erosion rates in their upstream catchments, which leads
to different slopes immediately above the point of confluence in the transport-limited model. As a consequence, the capture of
tributaries leads to stable knickpoints in the trunk stream for transport-limited erosion.

The most important lesson to be learned from this simple example concerns the estimation of the exponent $n$. Figure 5 shows
the relation between the steepness index $k_s$ and the erosion rate $E$, which is the same as the uplift rate $U$ in a steady state.
According to Eq. (14), the erosion rate is proportional to $k_s^n$ in the detachment-limited regime. So comparing $k_s$ at different
locations exposed to different erosion is a common approach to estimate $n$ (e.g., Lague, 2014).

The detachment-limited model with $K = 1$ and $n = 1$ reproduces the expected relation $E = k_s$, while this is not the case for the transport-limited model. Here the curve for the trunk stream (blue line) rather looks like a straight line with an offset. If we analyzed this curve without knowing that it originated from the transport-limited model, we would find that erosion starts at a threshold steepness index $k_s \approx 0.7$. If only a few points from this line were available, we would arrive at a nonlinear relation with $n > 1$. This is, however, an extreme example as $E = U = 0$ at the boundary, while the sediment flux from the domain requires a nonzero channel steepness. Qualitatively, the result would be similar in any situation where the uplift rate increases in upstream direction. If we interpret a long transport-limited river profile in terms of detachment-limited erosion, the exponent $n$ would be systematically overestimated.

In turn, comparing the three tributaries that are predominantly oriented in east-west direction would yield an exponent close to the correct exponent $n = 1$ used here. The reason is that these tributaries are not subject to strong variations in uplift rate. All estimates reviewed by Lague (2014) suggest $n > 1$ except for one data set. This data set describes strike-parallel tributaries originally investigated by Kirby and Whipple (2001) where a re-analysis by Wobus et al. (2006) resulted in $n \approx 1$. This finding sheds new light on the apparent evidence for exponents $n > 1$ obtained from analyzing river profiles under non-uniform conditions. An unrecognized contribution of sediment transport may result in an overestimation of $n$ here. This problem makes estimating the effective values of $n$ even more difficult and deserves further consideration in the future.

## 5   Numerical accuracy

While fully implicit schemes for diffusion-type equations are unconditionally stable for arbitrary $\delta t$, their numerical error increases linearly with $\delta t$. This error is a systematic error in the sense that the response of the system to temporal changes (e.g., in uplift here) is always too slow.

In order to assess this error and to estimate a reasonable maximum $\delta t$, the response of the steady-state topography with unit uplift considered in Sect. 4 to changes in uplift rate is investigated in the following. Due to the linearity of the model, the response to changes in $U$ is also linear as long as the flow directions do not change, which is the case if the change in $U$ is sufficiently small. Technically, the uplift rate can simply be set to $U = 0$ at $t = 0$, and the simulation is performed without recomputing the flow directions.

Figure 6 shows the results for different values of $\delta t$ in terms of the total sediment flux $Q$ out of the domain or, more precisely, in terms of the change in this flux per change in uplift rate, $\frac{\delta Q}{\delta U}$. The numerical error is in the order of magnitude of some percent for $\delta t = 1$. It should, however, be emphasized that this error is just some kind of time lag, while the sediment balance itself is satisfied exactly. The curves for $\delta t \leq 0.1$ are hardly distinguishable in the diagram. If we, e.g., assume $K = 2.5$ Myr$^{-1}$ (Robl et al., 2017), a unit of nondimensional time corresponds to 400,000 yr. So time increments in the order of magnitude of 100,000 yr should be no problem concerning the accuracy of the implicit scheme even for large grids ($5000 \times 5000$ nodes in this example).

While the numerical error of the implicit scheme is even smaller than for the detachment-limited model (Fig. 6), it must be taken into account that all models in this field compute flow directions and changes in topography in separate steps. Changes

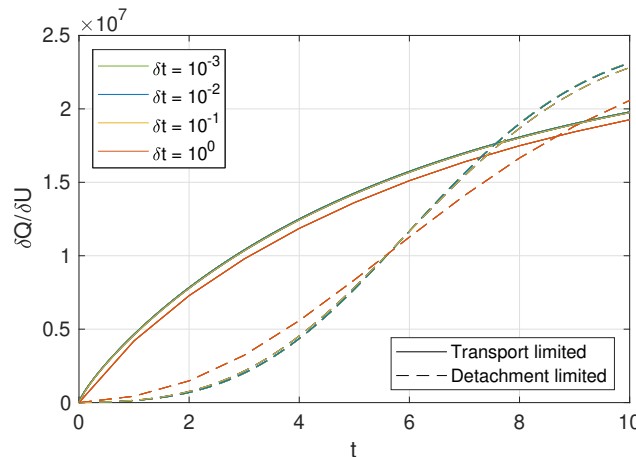

**Figure 6.** Change in sediment flux per change in uplift rate for different time increments $\delta t$ obtained from the steady-state topography with uniform uplift considered in Sect. 4.

in the flow directions introduce an additional numerical error. In the worst case, the implicit scheme acts on a deprecated flow pattern over almost the entire time step. So this error is formally also linear in $\delta t$, but strongly depends on the topography. Regions with small relief are particularly susceptible to artifacts. Under erosion, unreasonable river networks may deeply incise at large $\delta t$, and reorganization may take a long time afterwards. In the aggradational regime, large rivers may even turn into weird ridges within a single, large time step.

In many situations, the limitation of the maximum $\delta t$ arising from changes in the flow directions is more severe than the numerical error of the implicit scheme itself. However, as it strongly depends on the topography, it is difficult to provide an estimate for a reasonable $\delta t$ then. Practically, tracking the number of changes in flow direction and adjusting $\delta t$ so that the number of changes per time step does not exceed a given threshold provides a feasible criterion.

## 6    Extension towards the linear decline model

Detachment-limited erosion and transport-limited erosion can be seen as end members of a more general framework. In particular, the extension of the detachment-limited model by sediment transport proposed by Davy and Lague (2009) is receiving growing interest in this context. Recently, Guerit et al. (2019) derived estimates of the sediment deposition parameter occurring in this model from analyzing natural and experimental topographies. The authors concluded that "natural landscapes seem to describe a continuum between the two modes with a preference for TL mode" (transport-limited mode) as already suggested

by Davy and Lague (2009).

   Whipple and Tucker (2002) already proposed the generic form of the model of Davy and Lague (2009) and coined the term "linear decline model". This concept starts from the detachment-limited model and assumes that the sediment flux reduces the

ability of the river to erode. Assuming that the decrease in erosion rate is linear, this leads to the expression

$$E = f(A^\theta S) - \psi(A)Q \tag{28}$$

(from Eq. 4), where $\psi$ is an arbitrary function.

In addition to Eq. (28), the sediment balance (Eq. 2 or the respective discrete form, Eq. 12) must be satisfied. Inserting Eq. (28) and the sediment balance into Eq. (1) yields a system of two coupled partial differential equations for the surface height $H$ and the sediment flux $Q$.

The sediment balance can be written conveniently in integral form (Eq. 9). Combining this expression with Eqs. (1) and (28)

yields a single integro-differential equation for the surface height,

$$\frac{\partial H}{\partial t} = U - f(A^\theta S) + \psi(A) \int \left( U - \frac{\partial H}{\partial t} \right) dA \tag{29}$$

instead of a system of two differential equations.

The detachment-limited end member corresponds to $\psi(A) = 0$. In this case, the two differential equations are decoupled, so the equation for $H$ can be solved with computing $Q$. Approaching the detachment-limited end member is mathematically more

complicated. This can be achieved by increasing $f$ and $\psi$ in such a way that $f \to \infty$ and $\psi \to \infty$, while the ratio $\frac{f}{\psi}$ converges to a finite, nonzero value. Then, Eq. (28) turns into

$$Q = \frac{f(A^\theta S)}{\psi(A)}. \tag{30}$$

The resulting sediment flux $Q$ defines the transport capacity.

If we request that equilibrium river profiles under uniform conditions are still consistent with Hack's findings (Eq. 3), the

360 entire erosion rate defined by Eq. (28) must be a function of the product $A^\theta S$. Inserting Eq. (10) into Eq. (28) yields

$$E = \frac{f(A^\theta S)}{1 + A\psi(A)}. \tag{31}$$

So $\psi(A)$ must be inversely proportional to $A$,

$$\psi(A) = \frac{G}{A}, \tag{32}$$

with a nondimensional constant $G$. This is exactly the relation proposed by Davy and Lague (2009) (with $\Theta$ instead of $G$ there).

The second term in Eq. (28) turns into $\frac{GQ}{A}$, which was interpreted as deposition of sediments by Davy and Lague (2009).

The equilibrium erosion rate under uniform conditions is

$$E = \frac{f(A^\theta S)}{1 + G} = \frac{KA^m S^n}{1 + G} \tag{33}$$

in this model. So sediment deposition effectively reduces the erosion rate by a factor of $1 + G$ under uniform equilibrium conditions, which makes equilibrium profiles by a factor of $1 + G$ steeper in the linear model ($n = 1$) as already stated by Yuan

et al. (2019). This effect can be compensated by rescaling the erodibility $K$ by a factor of $1+G$, which modifies the model of Davy and Lague (2009) to

$$E = (1+G)\,KA^mS^n - G\frac{Q}{A}.$$

(34)

Both versions differ concerning the interpretation of the erodibility $K$. While it characterizes the process of detachment in the original model, it is interpreted as the fingerprint of spatially uniform erosion including sediment transport in the rescaled version. In contrast to the original version, the rescaled version also captures the transport-limited end member for $G \to \infty$ since

$$Q = \frac{(1+G)\,KA^mS^n - E}{\frac{G}{A}} \to KA^{m+1}S^n.$$

(35)

The linear decline model can be interpreted in several ways. If we define an equilibrium sediment flux by

$$Q_{\mathrm{e}} = \frac{f(A^\theta S)}{\psi(A)},$$

(36)

Eq. (28) turns into

$$E = \psi(A)\,(Q_{\mathrm{e}} - Q).$$

(37)

This is the undercapacity model (Kooi and Beaumont, 1994) written in terms of sediment flux instead of flux per unit width (Eq. 8).

The formulation of transport-limited erosion proposed in Sect. 2 allows for an alternative definition of a hybrid model that can also be interpreted as a linear decline model. Let us write the detachment-limited end member (Eq. 6) in the form

$$\frac{E}{K_{\mathrm{d}}} = A^mS^n,$$

(38)

and the transport-limited end member (Eq. 11) in the form

$$\frac{Q}{K_{\mathrm{t}}A} = A^mS^n.$$

(39)

In contrast to the previous considerations, different symbols $K_{\mathrm{d}}$ and $K_{\mathrm{t}}$ are used here. As discussed above, their meaning is in principle the same, but there is no reason why the values should be the same under all conditions.

The simplest combination of both end members is assuming that the property $A^mS^n$ that is responsible for both detachment and transport is shared among the two processes, i.e.,

$$\frac{E}{K_{\mathrm{d}}} + \frac{Q}{K_{\mathrm{t}}A} = A^mS^n.$$

(40)

This model approaches the detachment-limited regime for zero sediment flux and the transport-limited regime for high sediment flux.

Equation (40) can also be written in the form

$$E = K_{\mathrm{d}}A^mS^n - \frac{K_{\mathrm{d}}}{K_{\mathrm{t}}}\frac{Q}{A},$$

(41)

so

$$\psi(A) = \frac{K_\mathrm{d}}{K_\mathrm{t}A} \tag{42}$$

and

$$G = \frac{K_\mathrm{d}}{K_\mathrm{t}}. \tag{43}$$

The formulation defined by Eq. (40) could be called "shared stream power model". Compared to the concepts of detachment and deposition and the undercapacity model, is is rather a generic model. In turn, the formulation in terms of $K_\mathrm{d}$ and $K_\mathrm{t}$ may help to understand rivers passing different lithologies. Here we could expect that $K_\mathrm{d}$ shows a stronger variation than $K_\mathrm{t}$, although the ability to transport material also depends on the characteristics of the sediments at the river bed.

The numerical scheme described in Sect. 3 can be extended towards the linear version of the linear decline model, i.e., if the first term in Eq. (28) is also linear in channel slope $S$ ($n = 1$ in Eq. 6). The general form of this model reads

$$E = \phi(A)S - \psi(A)Q \tag{44}$$

with any functions $\phi$ and $\psi$. For the rescaled version of the model proposed by Davy and Lague (2009) and the shared stream power version, the two functions are

$$\phi(A) = \quad (1+G)\,KA^m \quad = K_\mathrm{d}A^m \tag{45}$$
$$\psi(A) = \quad \frac{G}{A} \quad = \frac{K_\mathrm{d}}{K_\mathrm{t}A}, \tag{46}$$

while the term $1 + G$ would not occur in the orginal version.

Inserting Eq. (44) into the general landform evolution model (Eq. 1) yields

$$\frac{\partial H_i}{\partial t} + \phi_i S_i - \psi_i Q_i - U_i = 0, \tag{47}$$

and after inserting difference quotients for time derivative and channel slope

$$\frac{H_i(t) - H_i(t_0)}{\delta t} + \phi_i \frac{H_i(t) - H_b(t)}{d_i} - \psi_i Q_i(t) - U_i = 0. \tag{48}$$

This equation can be rearranged in the form

$$H_i(t) = \frac{H_i(t_0) + \delta t\left(\frac{\phi_i}{d_i}H_b(t) + \psi_i Q_i + U_i\right)}{1 + \frac{\delta t \phi_i}{d_i}}. \tag{49}$$

Plugging this result into Eq. (19), rearranging the resulting equation to yield $Q_i(t)$, and comparing the obtained expression to Eq. (17) finally yields

$$Q_i^0 = \frac{\alpha_i\left(\frac{\phi_i}{d_i}\left(H_i(t_0) - H_b(t_0)\right) - U_i\right) + \beta_i\left(1 + \frac{\delta t \phi_i}{d_i}\right)}{\alpha_i \psi_i + 1 + \frac{\delta t \phi_i}{d_i}} \tag{50}$$

and

$$Q_i' = -\frac{\alpha_i \frac{\phi_i}{d_i}}{\alpha_i \psi_i + 1 + \frac{\delta t \phi_i}{d_i}}. \tag{51}$$

The scheme is very similar to that presented in Sect. 3 for transport-limited erosion. Equations (50) and (51) have to be used in sweep 2. Sweep 3 is now based on Eq. (49), while Eq. (17) is still used in its original form.

Preliminary numerical tests revealed that the time complexity of this version is very close to the transport-limited case, while that of the iterative scheme proposed by Yuan et al. (2019) is close to the detachment-limited case in each iteration step. In the first iteration, sweep 2 computes the catchment sizes here, while it integrates the upstream erosion rate to yield the sediment flux (Eq. 9) in subsequent iterations. Taking the values from Table 1, there would be a slight advantage of the iterative scheme (100 % vs. 112 %) if the iterative scheme could be applied with a single iteration step. This is, however, not possible if the flow direction of any node has changed because the sediment flux $Q$ is not available for the actual flow pattern then. In the best case, the iterative scheme requires two steps. According to the numerical tests of Yuan et al. (2019), this is achieved for small values of $G$ in the order of magnitude of 0.01 at $n = 1$. This yields 112 % vs. 162 % effort, so the direct scheme is at least 30 % faster than the iterative procedure. The advantage of the direct scheme rapidly grows with increasing sediment transport. The data repository of the recent study of Guerit et al. (2019) found a median value of $G = 1.6$ for $n = 1$ by analyzing several natural river profiles. The iterative scheme requires about 8 iterations at this value, resulting in an effort of 112 % vs. 534 %. So the direct scheme is almost 5 times faster under these conditions. Guerit et al. (2019) also reported on higher values of $G$, where the convergence of the iterative scheme would become very slow. In addition, the direct scheme has the advantage of an exact solution without the need for checking convergence.

## 7 Further extensions

### 7.1 Adding transport-limited and detachment-limited erosion

The models of the linear decline type discussed in Sect. 6 enforce a strict balance for the sediment flux for any nonzero function $\psi(A)$. However, we could also assume that a part of the eroded material is immediately excavated, while the rest is transported. This could be seen as a first step towards considering different particle sizes where one class of particles it so fine-grained that these will not be deposited. For simplicity, this version is elaborated only as an extension of the transport-limited model here, although a combination with the linear decline model is also possible. The linear version of this model reads

$$E = \mathrm{div}\boldsymbol{q} + \Gamma S, \tag{52}$$

or inserted into Eq. (1) and discretized in fully implicit form

$$s_i \frac{H_i(t) - H_i(t_0)}{\delta t} = s_i U_i - Q_i(t) + \sum_j Q_j(t) - s_i \Gamma_i S_i. \tag{53}$$

Here, $\Gamma$ is any function that describes the excavation of material. Similarly to the functions $\phi$ and $\psi$ used in the previous section, $\Gamma$ may in principle depend on all properties except for surface heights in order to maintain the linearity. It may be tempting to

use $\Gamma = \tilde{K} A^m$ in analogy to the detachment-limited model, where $\tilde{K}$ has the same meaning and physical unit as $K$. However, Eq. (52) combines a sediment balance with immediate excavation, which causes a scaling problem if rivers are considered as linear objects (Howard, 1994; Perron et al., 2008; Pelletier, 2010; Hergarten, 2020a). As a consequence, an additional rescaling factor depending on the pixel size must be introduced in the definition of $\Gamma$ in order to avoid an artificial dependence of the results on the spatial resolution. Different approaches for this scaling factor are discussed in the above references.

Apart from this scaling problem, the numerical implementation is straightforward. Using Eq. (11), the last term in Eq. (53) can be expressed as

$$s_i \Gamma_i S_i = \frac{s_i \Gamma}{K A_i^{m+1}} Q_i(t). \tag{54}$$

This results in a factor $1 + \frac{s_i \Gamma}{K A_i^{m+1}}$ in front of $Q_i(r)$ in Eq. (19). This factor propagates to the denominator of Eqs. (26) and (27), so that we finally arrive at

$$Q_i^0 = \frac{\alpha_i \left( H_i(t_0) - H_b(t_0) \right) + \beta_i \delta t}{\alpha_i \frac{d_i}{K A_i^{m+1}} + \left( 1 + \frac{s_i \Gamma}{K A_i^{m+1}} \right) \delta t} \tag{55}$$

and

$$Q_i' = -\frac{\alpha_i}{\alpha_i \frac{d_i}{K A_i^{m+1}} + \left( 1 + \frac{s_i \Gamma}{K A_i^{m+1}} \right) \delta t}. \tag{56}$$

The further steps (Eqs. 17 and 23) remain the same.

## 7.2 Hillslope diffusion

Linear diffusion (e.g., Culling, 1960) as the simplest model of hillslope erosion can be implemented more efficiently than in the detachment-limited model because the flux components in direction of the channel network can be integrated into the implicit scheme. If $d_{ij}$ is the distance between the node $i$ and a neighbor $j$ and $l_{ij}$ the length of the respective edge in a finite-volume representation, the diffusive flux in this direction is

$$Q_{ij}^{\text{diff}} = D l_{ij} \frac{H_i - H_j}{d_{ij}}, \tag{57}$$

where $D$ is the diffusivity. In contrast to the discrete divergence of the fluvial sediment flux density (Eq. 12), this simple expression is only valid if the edge is normal to the connecting line as it is the case, e.g., in a Voronoi discretization.

An implicit scheme for the flux components in direction of the channel network combined with an explicit scheme for the other components requires an additional variable $B_i$ (where practically either $Q_i^0$ or $Q_i'$ might be used) for the balance of the diffusive fluxes. For each node $i$, a loop over all neighbors $j$ with $H_j < H_i$ except for the flow target $b$ is employed. The respective values $Q_{ij}$ are added to $B_j$ and subtracted from $B_i$. After considering all nodes $i$, the values $\frac{B_i}{s_i}$ are added to the uplift rates $U_i$. This part of the scheme captures the diffusive fluxes except for those in direction of the channel network. Comparing Eq. (57) to Eq. (22), it is easily recognized that the diffusive flux from each node $i$ to its flow target $b$ can be included by replacing the term $K A_i^{\theta+1}$ by $K A_i^{\theta+1} + D l_{ib}$ throughout the calculations of Sect. 3.

As this scheme is not fully implicit, the maximum time increment is still limited. However, as the flux component in flow direction is the largest among all, its partly implicit treatment improves the stability of the diffusion term and thus increases the maximum possible time increment.

## 8 Limitations

While the approach presented here is efficient and can be applied to a large class of problems, some limitations should also be mentioned.

First, any kind of sediment transport that transfers material from one site to more than one target site destroys the tree-like topology of sediment fluxes. Such processes are thus not compatible with the implicit scheme presented here. This applies to hillslope processes as well as to fluvial processes with multiple flow directions as implemented, e.g., in the model TTLEM (Campforts et al., 2017). However, the implicit scheme for detachment-limited erosion is subject to the same limitation.

Concerning numerics, nonlinearity is the only point where the approach suggested here falls behind the implicit scheme for detachment-limited erosion. The latter can be solved directly for $n = 1$ and for $n = 2$ (Hergarten, 2002), but can be treated by finding the roots of a scalar nonlinear equation at each point for any value of $n$. In contrast, nonlinearity can only be included in the approach proposed here either by treating the nonlinear terms in an explicit manner or to employ an iteration.

Finally, the treatment of lakes, i.e., local depressions in the topography, is a problem. In the detachment-limited model, local depressions result in negative channel slopes and thus in negative erosion rates without any specific treatment. However, these negative erosion rates can be cut off easily in the implicit scheme. In the transport-limited model, local depressions result in a sediment flux opposite to the flow direction. Erosion of dams may be too fast then, so that the lifetime of lakes may be too short. This effect cannot be fixed easily in the fully implicit scheme.

## 9 Conclusions

This study proposes a simple formulation of transport-limited fluvial erosion. This formulation can be immediately reconciled with the empirical results of Hack (1957) on longitudinal river profiles. The interpretation of Hack's findings as the fingerprint of spatially uniform erosion is equivalent for transport-limited erosion and for detachment-limited erosion where it has been widely used. In turn, the behavior of both models differs if erosion is non-uniform.

As a main point, a new numerical scheme for treating transport-limited erosion with a fully implicit discretization in time was presented. It is a direct solver without any iteration and is unconditionally stable for arbitrarily large time increments. It is of linear time complexity ($O(N)$) where the computing effort is marginally higher than for detachment-limited erosion. The scheme can also be applied to combined linear models of detachment-limited erosion and sediment transport such as the linear decline model. Here it also allows for approaching the transport-limited end member without any loss of performance and provides a numerical efficiency that is better than the iterative scheme suggested by Yuan et al. (2019).

*Code and data availability.* All codes and computed data can be downloaded from the FreiDok data repository (Hergarten, 2020b). The author is happy to assist interested readers in reproducing the results and performing subsequent research.

*Author contributions.* N/A

*Competing interests.* The author declares that there is no conflict of interest.

The author would like to thank Xiaoping Yuan, Wolfgang Schwanghart, Jean Braun, and an anonymous reviewer for their constructive and encouraging comments. The article processing charge was funded by the Baden-Wuerttemberg Ministry of Science, Research and Art and the University of Freiburg in the funding programme Open Access Publishing.

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
