# Peer review of "Correspondence: Stefan Hergarten (stefan.hergarten@geologie.uni-freiburg.de)"

_Earth Surface Dynamics, 2020_

## Referee Comment (RC1) · Xiaoping Yuan (Referee) · 13 Jul 2020

General comments

This study proposes a simple formulation of transport-limited fluvial erosion. In particular, a new numerical scheme for treating transport-limited erosion with an implicit discretization in time was presented. It is also of linear time complexity ($O(n)$), similar to the scheme to solve the detachment-limited erosion model (Braun and Willett, 2013, Geomorphology). The scheme does not require iteration and is numberically stable for large time increments. This work also uses two landscape evolution simulations subjected to the uniform uplift and non-uniform uplift scenarios to show the model behaviours between the proposed transport-limited model and detachment-limited model.

[Figure]

Interestingly, the scheme can also be applied to the model of fluvial erosion and sediment deposition proposed by Davy and Lague (2019, JGR-ES). The direct scheme is at least 30% faster than the iterative scheme proposed by Yuan et al. (2019, JGR-ES), and even faster if the iterative scheme needs >=3 iterations. From these new aspects, the manuscript is suitable for publishing in Earth Surface Dynamics. However, the manuscript has some shortcomings on methods that need to be addressed and which may require minor-substantial revision.

Assumption in the transport-limit erosion: The author has an assumption to obtain the sediment flux in equations (10) and (11), which are important for the later derivation of implicit and O(n) scheme of the transport-limit erosion model. Erosion rate KAmSn in equation (6) is the rate at the outlet of drainage area A. The equation (11) assumes that the sediment flux of the drainage area equals to KAmSn × A = KAm+1Sn, which implies that the erosion rate is same everywhere in the drainage area A, which is true at the steady state subjected to a uniform uplift rate. Because at steady state, erosion everywhere balances rock uplift rate such that under conditions of spatially uniform uplift the total sediment flux at a given point along a river equals the product of upstream drainage area (A) and the rock uplift rate or the uniform erosion rate. This has been proved by the author using the uniform uplift rate (Figure 1, left panel) that transport-limited model produces the same, final steady-state landscape as the detachment-limited model. However, the author needs to show several transient-state comparisons between these two models before reaching steady state, and may test the sediment flux out of the domain to explore the differences between these two models (e.g., Armitage et al., 2018, ESurf). I have the feeling that they are different even they have the same final landscape.

Because of the above assumption (uniform erosion rate), in the case of non-uniform uplift rate (and thus non-uniform erosion rate at steady state), the transported-limit model produces different final landscape compared to the detachment-limited model (Figure 1, middle and right panels). The proposed transport-limit model, assumed a

uniform erosion, is unlikely suitable to study a non-steady-state (transient) landscape evolution or a non-uniform uplift scenario. Please argue against me if I am wrong.

The author mentioned that Yuan et al. (2019, JGR)'s erosion-deposition model/method breaks down if the model approaches the transport-limited regime, which is not true. Yuan et al. (2019) mentioned in their article that "..., the iterative method is proven to converge unconditionally at least when G ≤ 1, but we show experimentally that this method can also converge even if this condition is not satisfied", e.g., at G=10 (their Figure 3a), which is in transport-limited regime for G>1, a criteria estimated from various experimental and natural landscapes (Guerit et al., 2019, Geology).

I hope that these comments are helpful for the revision.

Xiaoping Yuan

Common and specific comments (by line)

L5: "as the stream-power law is" change to "of the stream-power law".

L7: "as the established implicit solver for transport-limited erosion", should be detachment-limited erosion?

L23 and L25: "sediment flux density" change to "sediment flux per unit width".

L67: A reference is needed for the upstream propagating velocity of erosion.

L72: "but despite increasing computing capacities still important point" change to "because increasing computing capacities is an important aspect in the landscape evolution modelling".

L78: Two 'n' in O(n) and S^n are confusing. Suggest to use 'O(N)' and N is the number of nodes discretizing the landscape. Suggest to change throughout the manuscript.

L134: Not easy to understand how to derive this equation (12) based on the above equations. Before the sentence, please write "Combine equations (1) and (12)", ....

L288-L291: It is better to list this computational time in Table 1.

References:

Armitage, J.J., Whittaker, A.C., Zakari, M. and Campforts, B., 2018. Numerical modelling of landscape and sediment flux response to precipitation rate change. Earth Surface Dynamics, 6(1), p.77.

Guerit, L., Yuan, X.P., Carretier, S., Bonnet, S., Rohais, S., Braun, J. and Rouby, D., 2019. Fluvial landscape evolution controlled by the sediment deposition coefficient: Estimation from experimental and natural landscapes. Geology, 47(9), 853-856.

---

## Referee Comment (RC2) · Wolfgang Schwanghart (Referee) · 14 Jul 2020

The manuscript by Stefan Hergarten details the derivation of an implicit solution for the transport-limited stream power model. An implicit scheme for the detachment-limited model was previously developed by the author (e.g., Hergarten and Neubauer 2001) and Braun and Willett (2013) and is popular in numerical landscape evolution models. An implicit solution for the stream power model with deposition was later developed by Yuan et al. (2019). However, the newly developed scheme is numerically more efficient and versatile.

Overall, the manuscript is very well written. It is easy to read and the mathematical development of the scheme is well explained. Moreover, the topic fits perfectly into

the scope of ESURF. I have no major concerns and only a few minor remarks. I thus recommend to accept the paper with only minor revisions.

Wolfgang Schwanghart

Minor comments:

164: S_i(t) is actually not used later, at least in the immediate context (Eqs. 20-25). You may, however, replace the second term in the right-hand side of Eq. 20 with S_i(t)

What about upstream boundary conditions? I wrote the model in MATLAB as 1-D model and the uppermost node remains fixed (if no uplift is applied). I may have wrongly written the code, though.

In the end, the model is not fully 2D, as the scheme is solved in 1D on a network. This may lead to weird aggradational forms (linear ridges on flat topography) if too long time steps are applied. Can you comment on this? What is an appropriate time step length?

References

Braun, J. and Willett, S. D.: A very efficient O(n), implicit and parallel method to solve the stream power equation governing fluvial incision and landscape evolution, Geomorphology, 180–181, 170–179, doi:10.1016/j.geomorph.2012.10.008, 2013.

Hergarten, S. and Neugebauer, H. J.: Self-Organized Critical Drainage Networks, Physical Review Letters, 86(12), 2689–2692, doi:10.1103/PhysRevLett.86.2689, 2001.

Yuan, X. P., Braun, J., Guerit, L., Rouby, D. and Cordonnier, G.: A New Efficient Method to Solve the Stream Power Law Model Taking Into Account Sediment Deposition, Journal of Geophysical Research: Earth Surface, 124(6), 1346–1365, doi:10.1029/2018JF004867, 2019.
* * *

---

## Author Comment (AC1) · 14 Jul 2020

Dear Xiaoping Yuan,

thanks for your constructive and encouraging comments! Although I have to argue against some of them, they will be very helpful for the revision.

First, however, I would like to point out that we apparently use slightly different terminologies concerning detachment-limited and transport-limited erosion. In my manuscript, I refer to the classical concept with detachment-limited and transport-limited erosion as end members that do, however, not occur in nature in this strict form. Anything in between would be called mixed channels then. In turn, you rather start from the approach suggested by Davy & Lague (2009) that is the basis of your

2019 paper. This approach already uses a quite specific (but probably very good) concept for the transition between the two end members, and then you distinguish the two regimes by a threshold parameter value (sediment deposition parameter $G = 0.4$). So you classify each scenario either as detachment-limited or transport-limited, while both are only the end members in my manuscript. The obviously led to some confusion.

As a main point of your review, you mention the difference in transient behavior between the detachment-limited and transport-limited models. "However, the author needs to show several transient-state comparisons between these two models before reaching steady state, and may test the sediment flux out of the domain to explore the differences between these two models (e.g., Armitage et al., 2018, ESurf). I have the feeling that they are different even they have the same final landscape." Yes, of course! Both end members have not much in common concerning their transient behavior. It is a second-order diffusion equation vs. a first-order advection equation. Transport-limited erosion in principle even supports no distinct transient knickpoints. I thought the difference in transient behavior was clear, and I just wanted to point out with the numerical example that they also differ under non-uniform steady-state conditions. The key point of Sect. 2 is just that the old findings of Hack (1957) could be alternatively be interpreted as transport-limited erosion in a uniform steady state.

Your second point refers to the suitability of the transport-limited model under inhomogeneous or under transient conditions. "The proposed transport-limit model, assumed a uniform erosion, is unlikely suitable to study a non-steady-state (transient) landscape evolution or a non-uniform uplift scenario. Please argue against me if I am wrong." Of course, the transport-limited end member is not suitable in bedrock mountain streams, and the detachment-limited end member not suitable for large parts of Earth's surface outside the mountain belts. We can conclude that we need combined models such as the one proposed by Davy & Lague (2009) in order to capture the majority of the rivers in the real world, but nothing more.

The third point addressing the convergence of your iterative scheme is just due to the

different terminology. "The author mentioned that Yuan et al. (2019, JGR)'s erosion-deposition model/method breaks down if the model approaches the transport-limited regime, which is not true." I referred to the transport-limited regime as an end member and not to $G > 1$. This end member is $G \to \infty$. The rate of convergence decreases with increasing $G$ and convergence indeed breaks down if we approach $G \to \infty$ according to my experience with the model. Mathematically, it is a singularly disturbed problem where the type of differential equation changes from hyperbolic to parabolic, and solving a parabolic equation iteratively with a scheme for a hyperbolic equation does not work well. This can of course be clarified in the revised version, but even the arguments in your own paper point towards avoiding too large values of $G$.

The specific point at the end of your reviews can be easily addressed in the revised version. Thanks for finding some mistakes!

Best regards,
Stefan Hergarten

---

## Author Comment (AC2) · 17 Jul 2020

Dear Wolfgang,

thanks for your encouraging comments! Just about the three minor points you mentioned:

*164: $S_i(t)$ is actually not used later, at least in the immediate context (Eqs. 20-25). You may, however, replace the second term in the right-hand side of Eq. 20 with $S_i(t)$.*

It is in fact used exactly the way you suggest since the direct form of Eq. (20) is $Q_i(t) = KA_i^{m+1}S_i$. But this is just the intermediate step from Eq. (11) to Eq. (20). In order to proceed to Eq. (21), however, I need the form of Eq. (20) used here. So the form of Eq. (20) with $S_i$ would be an intermediate step that might help the readers, but it cannot

be used as a simplification.

*What about upstream boundary conditions? I wrote the model in MATLAB as 1-D model and the uppermost node remains fixed (if no uplift is applied). I may have wrongly written the code, though.*

The "natural" upstream boundary condition is no influx of sediment (homogeneous von-Neumann), which is implicitly defined by the condition that the direction of the sediment flux follows the flow direction of water. So grid cells that are not supplied with water from other cells also do not receive sediments. In the equations, this means that all sums over the donors (all $\sum_j$ terms) are empty. Then $\alpha_i = s_i = A_i$ and $\beta_i = s_i U_i = A_i U_i$ (Eq. 18),

$$Q_i(t) = \frac{A_i \left(H_i(t_0) - H_b(t)\right) + A_i U_i \delta t}{A_i \frac{d_i}{K A_i^{m+1}} + \delta t}$$

(Eq. 23), and then into Eq. (21):

$$H_i(t) = H_b(t) + \frac{H_i(t_0) - H_b(t) + U_i \delta t}{1 + \delta t \frac{K A_i^m}{d_i}}.$$

This expression is even the same as the fully implicit step for the detachment-limited model, and according the formulation in Sect. 2 detachment-limited erosion and transport-limited erosion must indeed be the same at those sites without donors. So I guess that it might be just a problem in your 1D implementation.

*In the end, the model is not fully 2D, as the scheme is solved in 1D on a network. This may lead to weird aggradational forms (linear ridges on flat topography) if too long time steps are applied. Can you comment on this? What is an appropriate time step length?*

Yes, this problem affects all models where the computation of the flow pattern is separated from the change in topography. Changes in topography are then partly computed with the wrong flow pattern at large time step lengths. In the detachment-limited model,

it results in steep walls at drainage divides, and in the transport-limited model it generates these weird aggradational ridges. And as flow direction changes then, eroding them may even take much longer than generating them. However, it depends strongly on the considered situation. If a large river suddenly enters a completely flat area, it is definitely a problem, and it might even be necessary to use an adaptive scheme that also rejects time steps that resulted in a too large number of changes in flow direction. In turn, I am actually running large simulations of permanently changing rivers in a foreland with zero uplift, and there it seems not be a problem at all. So it is definitely an aspect that has to be taken into account, but I am actually not able to specify how strong it is under which conditions. Nevertheless it should be worth some words in the discussion part.

Best regards,
Stefan

---

## Referee Comment (RC3) · Anonymous Referee #3 · 18 Jul 2020

Review of Hergarten 2020 eSurf Transport-Limited

General comments:

This paper introduces a valuable new addition to the growing collection of efficient algorithms for computational landscape evolution models. Computational performance has long been a bottleneck for these models. Previous work by the author and by Braun and colleagues has yielded implicit $O(N)$ algorithms for wave-equation-like erosion laws, and more recently for the class of models that contain both erosion and sedimentation terms. Here Hergarten shows that slope-linear forms of the transport-limited and erosion-and-sedimentation models can be solved in $O(N)$ complexity using a direct implicit method. Bravo!

[Figure]

The extension to the erosion-and-sedimentation class of models makes the method broader than the title suggests, because that class of model can manifest a range of behavior across the detachment-limited to transport-limited spectrum. The author might therefore consider modifying the title to reflect this.

While it is true that Yuan et al. (2019) recently introduced an $O(N)$ algorithm for this class of models, there is certainly value in developing and comparing alternative numerical algorithms for the same problem. And appropriately, the author compares his new method with that of Yuan et al. (2019).

One aspect of the method that I find interesting is that the implicit algorithm should apply equally well for one-dimensional diffusion problems, provided that the flow in question is always oriented in one direction. That might not be tremendously helpful for people who want to model diffusion, since we already have numerous well-known solution methods in 1D and 2D for that particular problem, but it does suggest a way to test the proposed scheme under transient conditions. The manuscript notes that 'investigating the temporal behavior turned out to be quite complex'. Yet understanding temporal behavior is one of the reasons to use landscape evolution models in the first place. It is important to know something about the limits to accuracy and stability of a numerical scheme under transient conditions. I suggest therefore that the author try formulating a 1D, uni-directional diffusion problem and solving it with this implicit method for the transient case of a step change at one end of the domain. The analytical solution for that case is well known, so it seems like a good opportunity to test the properties of the proposed scheme under transient conditions. Basically, it would be a matter of having a 1D domain and a constant value of $A$.

Such a test might also make it possible to identify constraints on time-step size. The manuscript notes that implicit methods allow arbitrarily large time steps. Yes that's true in principle, but 'arbitrarily large' really just refers to stability. Two other considerations are: how does step size influence solution accuracy, and in particular for landscape models, to what extent does drainage network reorganization limit step size? These questions are undoubtedly hard or maybe impossible to answer in general, but some practical rules of thumb would be useful for those who wish to apply the algorithm in practice. So again I encourage the exploration of a transient case of simple 1D diffusion.

Comments by line number, equation, or figure:

the word 'uplift' has a long history of ambiguous usage among geoscientists. I recommend specifying 'uplift of crustal material relative to a given datum' or something like that.

I suggest adding some references here for the benefit of readers who are just getting into the topic. I am not sure of the provenance of the term 'transport limited', but I think it appears in Carson and Kirkby (1972) in the context of hillslopes. For the landscape evolution context, Willgoose et al. (1990, Water Resources Research) might be a reasonable reference, though I do not remember whether they actually used this phrase. As far as I know, the term 'detachment limited' was coined by Howard (1994, Water Resources Research).

Up to this point, you have not actually defined transport limited. This would be a good place to do so. I think of a transport-limited river reach as one in which the rate of bed erosion is limited by the ability of the flow to transport the eroded material downstream, rather than by the availability of potentially mobile sediment (feel free to use this wording if you like it).

For what it is worth, in my view, the definition is actually fuzzier than we sometimes pretend: the ability of moving fluid to transport sediment depends very strongly on the size and density of sediment on the bed. There is no such thing as a 'transport capacity' independent of bed sediment characteristics. Bed-load theory tells us that transport capacity depends on critical shear stress, which in turn depends on sediment size and density; suspended-load theory tells us that sediment concentration depends on near-bed sediment concentration and on settling velocity, both of which also depend on size and density. But for purposes of this paper, the only real practical implication of this observation is that one should be cautious in using the phrase 'transport capacity.'

31-33 Second-order derivatives only appear if $q$ is a function of topographic gradient. Suggest adding wording to clarify this, e.g., 'Because $q$ is a function of topographic gradient, eq (1) contains...'

Change 'In the last years' to 'In recent years'

'there seems to be a trend to'—I also share the impression that use of a detachment-limited stream erosion model in landscape evolution studies is common, but whether there has been a trend in that direction is harder to say. There are situations in which a transport-limited model is suitable, and plenty of literature on such models (e.g., Wickert and Schildgen, 2019; and a great deal of the work by Greg Hancock, Tom Coulthard, and colleagues). Suggest simply asserting that detachment limited is a common or popular choice.

Of the three suggested reasons for the widespread use of detachment-limited discharge-slope models, I think the second two are really the important ones. The first might be a bit misleading to readers, because any of the three flavors of model discussed in this paper can be related to a power-law slope-area relationship. As far as I know, the link between erosion/transport and slope-area was actually first identified in a transport-limited context. If I recall right, Howard (1980 in Thresholds in Geomorphology) articulated a slope-area relation based on a variety of different transport formulas, and Howard and Kerby (1983) followed up with a field-based study. Then Willgoose et al. (1991) and Willgoose (1994) really hit home the slope-area relation in a transport-limited context. So having a link with Flint's law is not unique to the detachment-limited formulation. The solution I suggest is just to add a sentence, maybe after the sentence following eq (6), to the effect that 'transport-limited and other types of erosion law can also be linked to Flint's law (references), but the relationship is especially simple for the area-slope erosion law in eq (6)'.

'has become some kind of paradigm'—I think I understand what you mean here, but as written it is a vague statement (what exactly constitutes a paradigm? what 'kind' of paradigm?). Better I think to leave this comment out.

'little is known'—this statement is a bit unfair to researchers who have tried to pin it down. Suggest softening to something like 'the effective value of $n$ is less well known'. You could also add something like: 'some studies suggest a linear scaling (REFS), some sub-linear (REFS), and some super-linear (REFS)'.

There are quite a few other papers that report estimates of $K$ values, which could be cited here. I guess the 'e.g.' is meant to say 'there are more papers than I feel like bothering to list here, but if you want a starting point, try these two'. I guess that's ok, but you are likely to annoy the authors of the ones you left out. An alternative would be to find a recent paper or two that reports $K$ values and is reasonably comprehensive in its referencing, and cite as 'So-and-so, 20xx, and references therein'.

The wording is a bit awkward here; suggest leaving out 'despite increasing computing...' (we all know computers have gotten faster).

'models treat'

confusing because you would choose either (2) or (8); how about '(1) and either (2) or (8)'

the upstream eq (11) and preceding text: the way this is written seems to suggest that approaching the problem from the question of 'how much sediment would you get from eq (6)' is a requisite for deriving the method that follows. Actually, there are at least two other pathways that I can think of. I think you are more likely to 'sell' the approach more effectively if you point out that there are several lines of evidence to support the hypothesis that the long-term sediment flux should depend on slope and drainage area. You have articulated one of them, but it seems to me it is subject to the criticism that you are using a detachment-limited concept (eq 6) to derive a transport-limited model. An alternative would be to state that previous studies have shown that sediment-transport formulas can be cast in the form of an area-slope power expression, and cite some references. You could also lean on Davy and Lague (2009) here, because when you combine their expression with a unit-stream-power detachment rate and $Q_w \propto A$ ($Q_w$ being water discharge), you end up with a transport capacity (if I recall right) that looks like $A^{3/2}S$ (more generally, $A^{m+1}S^n$). I think it is fair to say that there is uncertainty in the literature over how best to express transport capacity in models of stream profile evolution or landscape evolution. Some have $Q \propto Q_w S$, some (e.g., Willgoose, Howard, based on the empirical Einstein-Brown expression) have $Q \propto q_w^2 S^2$, and some include a transport threshold. Key point for your purposes is that $Q = KA^{m+1}S$ falls within the span of proposed laws.

change which to that (introduces a restrictive clause)

132-3 I do not understand this comment about Voronoi polygons. Normally in a finite-volume solution, you would integrate flux density over the width of a cell face, whether it is a square or a Voronoi polygon or some other shape. Using $Q$ instead of $q$, with an implied sub-grid-scale channel width (I suppose), you do not need to do this integration; but that is true regardless of the shape of your cells. Is your point that the discrete representation in eq 12 works in principle for any grid mesh, regular or irregular? Consider removing this statement, as it seems like a bit of a distraction.

137-8 See comment above about transport laws. Equations like (11) have been frequently used in the literature. In particular, in the work of Willgoose et al. (1991a,b,c) and subsequently, the slope-area relationship is used to estimate parameters for a transport law. If there is something very specific about eq (11) that you think is unique, then that should be pointed out. Otherwise, the statement carries the implication that transport laws have never before been derived from slope-area analysis, which is not correct.

[Figure]

144-5 I believe it is more than a matter of terminology. It is rather a matter of dimensionality. If you suppose $m = 1/2$, then the erodibility has dimensions of inverse time, whereas the transport coefficient has dimensions of length$^2$ / time.

153-157 Consider adding some more explanatory text here. At first glance, eq (15) looks like a Taylor expansion to first order. But if I am following this correctly, actually $Q_i^0$ includes the value of $Q_i$ at $t_0$ plus the partial derivative of $Q_i$ with respect to $H_i$ times the change in $H_i$ during one time step. That's a clever idea, and is consistent with the explanation on line 156, but it took me some time to work it out. Other readers might similarly mis-interpret $Q_i^0$ on a first look, and yet its definition is really key to whole scheme. Suggest devoting a full sentence or so to pointing out the definition and importance of it.

The challenge for readers is that the donor information is buried in the definitions of alpha and beta. Suggest adding, after the word donors, '(because $\alpha$ and $\beta$ depend on donors' $Q'$ and $Q$, respectively)'

187, 189 - reference to a recursive implementation is vague. Suggest referring to a published algorithm(s) for sorting by downstream order.

198-200 Please document somehow the specifications for the performance tests: for example, the number of iterations were run for each case.

207-8 With all due respect, I think this is a missed opportunity. As noted above, I suggest trying a solution with one row of grid nodes (so, strictly one dimensional) and a uniform drainage area. Then it reduces to a linear diffusion equation, which you could compare with the transient analytical solution for diffusion given a step change at one boundary.

Please explain the rationale for increasing $\delta t$ over time.

217-8 To avoid potential confusion, it would be useful to clarify that the two models are NOT equivalent, but rather their steady state solutions have the same slope-area relationship. Either give the predicted slope-area equivalence, or quote a reference that does (or both).

and following: The tent-shaped uplift pattern is a clever test. I think the example would be easier to follow if you did two things. First, before referring to the results (figure 2), explain why you are using this tent-shaped uplift and what differences you expect to see between the two models. That way, the reader knows what to look for in Figures 2 and 3. Second, it would be very helpful to provide an analytical solution for the two models. You could simply use Hack's law to relate drainage area to distance (I would just make the exponent 2 for simplicity). Plot the predicted longitudinal stream profiles with a tent-shaped uplift pattern for each model, in chi space (you could do linear space too). That way there is a clear expectation for Figure 3 (actually, you could simply add the analytical profiles to Figure 3). If I have done the math right, the two profiles should be defined by

$$dH/dx = (u_0(L-x)/K)x^{-hm}$$
$$dH/dx = (u_0/K)x^{-h(m+1)} \int_0^x (L-x)dx$$

where $u_0$ is the uplift rate at the ridgeline, $h$ is the Hack exponent, and $L$ is the domain half-width. So, should be possible to plot these as analytical expectations.

$\phi$ and $\psi$ seem to be parameters rather than functions.

Davy and Lague deserve much credit for introducing this formulation in the landscape evolution context, and showing that it relates to the earlier Beaumont model except that the length scale varies with unit discharge. For the record, similar formulations with erosion/entrainment and deposition terms seem to be widely used in the sedimentation engineering and soil erosion communities.

eq (34) I like this alternative expression of phi. Presumably it would simplify calibration by removing a built-in correlation between the two parameters.

284-5 Would not $G \to \infty$ lead to $Q_i' \to 0$ by eq (31)?

missing 'to'

extra 'of'

'some kind of' is a bit vague. Suggest re-wording to be more precise.

295-6 Can you articulate what process(es) this kind of formulation is meant to represent? Is the idea that some of the material is so fine-grained that it will not end up being deposited until it reaches the ocean or some kind of closed basin? I wonder whether an alternative would be to build this into $dQ/dH$.

303-4 This statement is not clear to me. From the references cited, I guess that by 'scaling problem' you mean the classic problem of grid-size scaling. Yet that wasn't mentioned as an issue with the prior models (transport limited and linear decline), so why is it more of an issue with equation 35 than with, say, equation 26?

317-325 Are you suggesting to solve for diffusive flux in the flow directions using the implicit scheme, and the other directions using some other scheme? How would you avoid double-counting the fluxes in the cardinal flow directions? Overall, I think the sketch presented here for handling diffusion is not really convincing. I would recommend either deleting it, or expanding it to really demonstrate how it would work.

Finally: nice work!

---

## Editor Comment (EC1) · Jean Braun (Editor) · 24 Jul 2020

This is a very interesting contribution for those of us working on developing efficient and stable solutions to the basic equations governing surface erosion and sediment transport and deposition. It describes an $O(n)$ complexity and time-implicit method to solve the transport-limited equation (i.e., where change in surface height is equal to the divergence of sediment flux). It is a very important contribution that certainly highly suitable for publication in ESURF. Like in most previous studies, the expression used for the flux ($Q = KA^{m+1}S^n$) is chosen such that the steady-state topography obtained is identical to that obtained assuming the stream power incision model (SPIM) where erosion rate is given by $KA^mS^n$. The author rightly reminds us that the transient behaviour of a "divergence"-based approach is however different from that obtained by

the SPIM. He also states that characterising this difference is not the purpose of this manuscript, and I agree with this.

The author also proposes to extent the method to solve another equation obtained by assuming that the rate of erosion/deposition, $E$, is the sum of an erosion term (assumed proportional to slope, $S$) plus a deposition term assumed proportional to sediment flux, $Q$ ($E = \phi S + \psi Q$), which leads to a hybrid behaviour that can reproduce the observed transition between detachment-limited behaviour of the system when sediment flux is low to transport-limited behaviour when sediment flux increases (as proposed by Kooi and Beaumont in 1994 and Davy and Lague in 2009). This, I find, is the most interesting part of the manuscript.

The limitations of the method are well explained such as the first-order representation of the slope which tends to affect the sharpness of knickpoints. It is also limited to specific cases (i.e. $n = 1$ and $n = 2$). The author also mentions the problems caused by local depressions/minima without suggesting a potential solution.

I think the author could provide a more quantitative comparison between the results obtained with this new method and those obtained by solving the Davy and Lague (2009) approach directly (i.e. without using the flux divergence formulation) as done by Yuan et al (2019). I have rapidly coded the two approaches in 1D and easily showed that the improvement in speed is a strong function of $G$ (for $G \geq 1$).

I found the paper well written and agree with the opinion expressed by reviewers (1 and 2) and with the minor modifications they suggest. Reviewer 3 makes more substantial and also very interesting suggestions for improvement to the paper. All agree that this is a very important contribution that should be published in ESurf.

Additionally to the suggestions made by the reviewers, I would like the author to provide a more structured explanation of the procedure used in the solution of what the author calls the "linear decline model". I note that the author's implementation is based on equations [1]+[26] (I use square brackets to indicate equation numbers from the

manuscript and parentheses to indicate equation numbers from this comment):

$$\frac{\partial H}{\partial t} = U - \phi S + \psi Q \tag{1}$$

and its discretised form [28]; but, of course, it also uses equation [17], which is equivalent to [16], itself a discretised form of equation:

$$\frac{\partial H}{\partial t} = U - \text{div}\mathbf{q} \tag{2}$$

This leads me to conclude that the author uses two evolution equations simultaneously at every point of the landscape. Combining them also suggests that:

$$\text{div}\mathbf{q} = \phi S - \psi Q \tag{3}$$

This may explain why the method is hybrid, depending on the value of the $\psi$ parameter (purely advective if $\psi = 0$ and divergence based (or diffusive) for large values of $psi$), as pointed out by the author. However, it is not clear to me how to connect all these points together. I would appreciate if the authors could help me (and other readers) clarify this point with a more structured presentation of the basic partial differential equation(s?) that are solved in the "linear decline model" algorithm.

---

## Author Comment (AC3) · 26 Jul 2020

Dear reviewer,

thanks for your encouraging comments and for the suggestions to improve the manuscript!

As the official discussion phase will end soon, I just write a few remarks on some points that might also be interesting for other readers and go more into detail when submitting the revised version.

**Time-dependent problems:** My statement "investigating the temporal behavior turned out to be quite complex" should not mean that I do not want to consider it. However, catching up with what has already been done for the detachment-limited case just

in one section of this paper is impossible. We know what solutions of the 1D diffusion equation look like, but the interesting aspect how disturbances propagate into tributaries where the diffusivity is lower than in the trunk stream. My recent Postdoc researcher has already started a study on the characteristic response time of catchments to changes in uplift or base level and which part the hillslopes play here. But this will be an own paper where I will not be first author.

**Time-step size:** You are right that the unlimited time-step size only refers to the situation with frozen flow directions. If a river enters a flat area, even weird aggradational ridges may occur and vanish after some time. Their size depends linearly on the time-step length, and we need very small time steps to avoid this problem if the river brings much sediments. In turn, I am actually running large simulations of permanently changing rivers in a foreland with zero uplift, and there it seems not be a problem at all. So it will even be a challenge to provide some rule of thumb.

**Voronoi polygons:** You are right that any finite-volume representation could be used. For the accuracy it is, of course, and advantage if the edge is in the middle of the nodes, and Voronoi polygons turn out to be useful if hillslope diffusion is considered additionally.

**Equation (15):** The linear dependence of all properties (height, flux) on base level is indeed the key to understanding the numerical scheme, so I agree that it should be described more thoroughly. I already prepared a figure where this is visible and where it can be recognized that it is not just a first-order Taylor approximation.

**Tent-shaped uplift pattern:** It would indeed be great to have analytical solutions (approximations) for the river profiles as you suggest. As far as I can see, your analytical solution is technically correct, but runs into problems with the contribution of tributaries for the transport-limited model. As an extreme example, all single-pixel catchments are located at the ridge and thus exposed to the maximum uplift in the 1D formulation using Hack's law. In the network, however, they are distributed over the entire catchment. So
they are exposed to lower uplift on average and thus provide less sediment. I recently worked on a similar problem in the context of glacial erosion (Prasicek et al., EPSL, 2020, 0.1016/j.epsl.2020.116350), and it is more complicated that it seems first.

**Extension by hillslope diffusion:** This is indeed not very complicated and works quite well at least for triangular meshes. I will explain the implementation in the revised version.

Best regards,
Stefan Hergarten

---

## Author Comment (AC4) · 27 Jul 2020

Dear editor,

thanks for your encouraging comments and for the suggestions to improve the manuscript! Im a quite happy that you also tested it in the 1-D case and compared the performance to the implementation of Yuan et al. (2019).

As the official discussion phase will end soon, I just write a few remarks on some points that might also be interesting for other readers and go more into detail when submitting the revised version.

**Performance:** You are right that the improvement in performance compared to the iterative implementation of Yuan et al. (2019) rapidly increases with increasing $G$ for

$G > 1$ as the convergence of the iterative scheme becomes slower. I will discuss it a bit more in detail based on the performance test provided by Yuan et al. (2019) in Fig. 2 in combination with the estimates of $G$ in natural rivers recently provided by Guerit et al. (2019).

**Linear decline model:** I agree that the extension towards hybrid models is the most interesting part for applications, and that that the concept behind this model should be explained more thoroughly. So I will devote an own section to this topic clarifying the structure of the differential equations (system of two partial differential equations for the height $H$ and the sediment flux $Q$ that collapses to a single equation for the detachment-limited and transport-limited end members), explaining the relation to the approaches of Kooi and Beaumont (1994) and Davy and Lague (2009), and providing a third interpretation in terms of shared stream power that may be useful when different lithologies are considered.

Best regards,
Stefan

---

## Author Response (AR1)

Dear Reviewers, dear Editor,

thanks for your constructive and encouraging comments! In the following, the points addressed in your reports are discussed, and changes to the manuscript are described. Line numbers refer to the version with highlighted changes at the end of this document.

**Reviewer 1 (Xiaoping Yuan)**

"Assumption in the transport-limit erosion: The author has an assumption to obtain the sediment flux in equations (10) and (11), which are important for the later derivation of implicit and O(n) scheme of the transport-limit erosion model. Erosion rate $KA^mS^n$ in equation (6) is the rate at the outlet of drainage area A. The equation (11) assumes that the sediment flux of the drainage area equals to $KA^mS^n \times A = KA^{m+1}S^n$, which implies that the erosion rate is same everywhere in the drainage area $A$, which is true at the steady state subjected to a uniform uplift rate. Because at steady state, erosion everywhere balances rock uplift rate such that under conditions of spatially uniform uplift the total sediment flux at a given point along a river equals the product of upstream drainage area $(A)$ and the rock uplift rate or the uniform erosion rate. This has been proved by the author using the uniform uplift rate (Figure 1, left panel) that transport-limited model produces the same, final steady-state landscape as the detachment-limited model. However, the author needs to show several transient-state comparisons between these two models before reaching steady state, and may test the sediment flux out of the domain to explore the differences between these two models (e.g., Armitage et al., 2018, ESurf). I have the feeling that they are different even they have the same final landscape."

Yes, of course! Both end members have not much in common concerning their transient behavior. It is a second-order diffusion equation vs. a first-order advection equation. Transport-limited erosion in principle even supports no distinct transient knickpoints. I thought the difference in transient behavior was clear, and I just wanted to point out with the numerical example that they also differ under non-uniform steady-state conditions. The key point of Sect. 2 is just that the old findings of Hack (1957) could be alternatively be interpreted as transport-limited erosion in a uniform steady state. **I pointed out this more clearly at the end of Sect. 2 (lines 178–186).**

"Because of the above assumption (uniform erosion rate), in the case of non-uniform uplift rate (and thus non-uniform erosion rate at steady state), the transported-limited model produces different final landscape compared to the detachment-limited model (Figure 1, middle and right panels). The proposed transport-limit model, assumed a uniform erosion, is unlikely suitable to study a non-steady-state (transient) landscape evolution or a non-uniform uplift scenario. Please argue against me if I am wrong.

Of course, the transport-limited end member is not suitable in bedrock mountain streams, and the detachment-limited end member not suitable for large parts of Earth's surface outside the mountain belts. We can conclude that we need combined models such as the one proposed by Davy & Lague (2009) in order to capture the majority of the rivers in the real world, but nothing more.

"The author mentioned that Yuan et al. (2019, JGR)s erosion-deposition model/method breaks down if the model approaches the transport-limited regime, which is not true. Yuan et al. (2019) mentioned in their article that '..., the iterative method is proven to converge unconditionally at least when $G \leq 1$, but we show experimentally that this method can also converge even if this condition is not satisfied', e.g., at $G = 10$ (their Figure 3a), which is in transport-limited regime for $G > 1$, a criteria estimated from various experimental and natural landscapes (Guerit et al., 2019, Geology)."

This is just due to slightly different terminologies. In my manuscript, I refer to the classical concept with detachment-limited and transport-limited erosion as end members that do, however, not occur in nature in this strict form. Anything in between would be called mixed channels then. In turn, you subdivide the entire range into a detachment-limited and a transport-limited range. **I clarified this by declaring detachment-limited erosion and transport-limited erosion as end members (lines 9, 17, 132, 134, 271, 272, 346, 364, 365, 387, 397, 399, 405, and 559).**

"L5: 'as the stream-power law is' change to 'of the stream-power law'."

I changed the wording of this sentence either **(lines 5–6)**.

"L7: 'as the established implicit solver for transport-limited erosion', should be detachment-limited erosion?"

Thanks! I could have read this 100 times without finding this mistake. **Fixed (line 8).**

"L23 and L25: 'sediment flux density' change to 'sediment flux per unit width'."

Good idea as flux density could be misinterpreted as flux per area. **I changed it throughout the paper (lines 26, 32, 115, 143, 151, 162, 359).**

"L67: A reference is needed for the upstream propagating velocity of erosion."

I would say it is too simple for a reference as it is just the velocity of advection in the advection equation. **I added a short explanation (line 84).**

"L72: 'but despite increasing computing capacities still important point' change to 'because increasing computing capacities is an important aspect in the landscape evolution modelling'."

**I removed this phrase because everybody knows that computing capacities have increased (line 94).**

"L78: Two '$n$ in $O(n)$ and $Sn$ are confusing. Suggest to use '$O(N)$ and $N$ is the number of nodes discretizing the landscape. Suggest to change throughout the manuscript."

**Good point; I changed it throughout the paper (lines 96, 97, 254, 557).**

"L134: Not easy to understand how to derive this equation (12) based on the above equations. Before the sentence, please write 'Combine equations (1) and (12)', ..."

Ok, although this is not the most difficult part of the paper. **I added the explanation (line 164).**

"L288-L291: It is better to list this computational time in Table 1."

Table 1 occurs much earlier in the paper. So I think that it would not be helpful to mention these values already there. Beyond this, it would make Table 1 more complicated as it refers to a single step so far.

**Reviewer 2 (Wolfgang Schwanghart)**

"164: $S_i(t)$ is actually not used later, at least in the immediate context (Eqs. 20-25). You may, however, replace the second term in the right-hand side of Eq. 20 with $S_i(t)$."

It is in fact used exactly the way you suggest since the direct form of Eq. (20) is $Q_i(t) = KA_i^{m+1}S_i$. But this is just the intermediate step from Eq. (11) to Eq. (20). In order to proceed to Eq. (21), however, I need the form of Eq. (20) used here. So the form of Eq. (20) with $S_i$ would be an intermediate step that might help the readers, but it cannot be used as a simplification.

"What about upstream boundary conditions? I wrote the model in MATLAB as 1-D model and the uppermost node remains fixed (if no uplift is applied). I may have wrongly written the code, though."

The "natural" upstream boundary condition is no influx of sediment (homogeneous von-Neumann), which is implicitly defined by the condition that the direction of the sediment flux follows the flow direction of water. So grid cells that are not supplied with water from other cells also do not receive sediments. In the equations, this means that all sums over the donors (all $\sum_j$ terms) are empty. Then $\alpha_i = s_i = A_i$ and $\beta_i = s_i U_i = A_i U_i$ (Eq. 18),

$$Q_i(t) = \frac{A_i\left(H_i(t_0) - H_b(t)\right) + A_i U_i \delta t}{A_i \frac{d_i}{KA_i^{m+1}} + \delta t}$$

(Eq. 23), and then into Eq. (21):

$$H_i(t) = H_b(t) + \frac{H_i(t_0) - H_b(t) + U_i \delta t}{1 + \delta t \frac{KA_i^m}{d_i}}.$$

This expression is even the same as the fully implicit step for the detachment-limited model, and according the formulation in Sect. 2 detachment-limited erosion and transport-limited erosion must indeed be the same at those sites without donors. So I guess that it might be just a problem in your 1D implementation. **I added a short node on the boundary conditions (lines 217–219)**.

"In the end, the model is not fully 2D, as the scheme is solved in 1D on a network. This may lead to weird aggradational forms (linear ridges on flat topography) if too long time steps are applied. Can you comment on this? What is an appropriate time step length?"

Yes, this problem affects all models where the computation of the flow pattern is separated from the change in topography, so also the detachment-limited model. How severe it is and how much it limits the maximum $\delta t$, depends strongly on the considered situation. So it is even difficult to provide a rule of thumb for the maximum $\delta t$. **I added some remarks in the section about the limitations (lines 535–543)**.

**Reviewer 3**

"One aspect of the method that I find interesting is that the implicit algorithm should apply equally well for one-dimensional diffusion problems, provided that the flow in question is always oriented in one direction. That might not be tremendously helpful for people who want to model diffusion, since we already have numerous well-known solution methods in 1D and 2D for that particular problem, but it does suggest a way to test the proposed scheme under transient conditions. The manuscript notes that investigating the temporal behavior turned out to be quite complex. Yet understanding temporal behavior is one of the reasons to use landscape evolution models in the first place. It is important to know something about the limits to accuracy and stability of a numerical scheme under transient conditions. I suggest therefore that the author try formulating a 1D, uni-directional diffusion problem and solving it with this implicit method for the transient case of a step change at one end of the domain. The analytical solution for that case is well known, so it seems like a good opportunity to test the properties of the proposed scheme under transient conditions. Basically, it would be a matter of having a 1D domain and a constant value of A."

"Such a test might also make it possible to identify constraints on time-step size. The manuscript notes that implicit methods allow arbitrarily large time steps. Yes thats true in principle, but arbitrarily large really just refers to stability. Two other considerations are: how does step size influence solution accuracy, and in particular for landscape models, to what extent does drainage network reorganization limit step size? These questions are undoubtedly hard or maybe impossible to answer in general, but some practical rules of thumb would be useful for those who wish to apply the algorithm in practice. So again I encourage the exploration of a transient case of simple 1D diffusion."

Transient behavior is, of course, the heart of landform evolution modeling. However, catching up with what has already been done for the detachment-limited case just in one section of this paper is impossible. We know what solutions of the 1D diffusion equation look like, but the interesting aspect how disturbances propagate into tributaries where the diffusivity is lower than in the trunk stream. My recent Postdoc researcher has already started a study on the characteristic response time of catchments to changes in uplift or base level and which part the hillslopes play here. But this will be an own paper where I will not be first author.

Formally, the error of the scheme is linear in $\delta t$, but this does not help much practically. The reorganization of the drainage network is indeed the limiting factor here. This problem affects all models where the computation of the flow pattern is separated from the change in topography, so also the detachment-limited model. How severe it is and how much it limits the maximum $\delta t$, depends strongly on the considered situation. So it is even difficult to provide a rule of thumb for the maximum $\delta t$. **I added some remarks in the section about the limitations (lines 535–543).**

"14 the word uplift has a long history of ambiguous usage among geoscientists. I recommend specifying uplift of crustal material relative to a given datum or something like that."

"16 I suggest adding some references here for the benefit of readers who are just getting into the topic. I am not sure of the provenance of the term transport limited, but I think it appears in Carson and Kirkby (1972) in the context of hillslopes. For the landscape evolution context, Willgoose et al. (1990, Water Resources Research) might be a reasonable reference, though I do not remember whether they actually used this phrase. As far as I know, the term detachment limited was coined by Howard (1994, Water Resources Research)."

"26 Up to this point, you have not actually defined transport limited. This would be a good place to do so. I think of a transport-limited river reach as one in which the rate of bed erosion is limited by the ability of the flow to transport the eroded material downstream, rather than by the availability of potentially mobile sediment (feel free to use this wording if you like it)."

"For what it is worth, in my view, the definition is actually fuzzier than we sometimes pretend: the ability of moving fluid to transport sediment depends very strongly on the size and density of sediment on the bed. There is no such thing as a transport capacity independent of bed sediment characteristics. Bed-load theory tells us that transport capacity depends on critical shear stress, which in turn depends on sediment size and density; suspended-load theory tells us that sediment concentration depends on near-bed sediment concentration and on settling velocity, both of which also depend on size and density. But for purposes of this paper, the only real practical implication of this observation is that one should be cautious in using the phrase transport capacity. "

I think that it would not be a big problem for the readers in combination with the equation. Nevertheless, the is no argument against clarifying it, so **I added the suggested phrase (line 15)**.

Admittedly, I am completely uncertain about the provenance of the two terms. I agree that it makes sense to mention the two papers from the early 1990s where the two concepts presumably occurred for the first term in the context of these types of models, so **I added the references (lines 19 and 30–31)**.

It did even not think that the term 'transport limited' requires a definition, but it makes sense and I like your suggested wording, so **I added it (lines 29–30)**.

I fully agree. The consequence is that the parameter $K$ in my formulation is a lumped parameter that also depends on the characteristics of the material coming from the upstream area. **A little remark in this direction now occurs in line 417–419**.

"31-33 Second-order derivatives only appear if q is a function of topographic gradient. Suggest adding wording to clarify this, e.g., Because q is a function of topographic gradient, eq (1) contains... "

**I took it out of the parentheses and explained it a bit more clearly now (lines 39–40).**

"35 Change In the last years to In recent years "

**Corrected (line 43).**

"36 there seems to be a trend to – I also share the impression that use of a detachment-limited stream erosion model in landscape evolution studies is common, but whether there has been a trend in that direction is harder to say. There are situations in which a transport-limited model is suitable, and plenty of literature on such models (e.g., Wickert and Schildgen, 2019; and a great deal of the work by Greg Hancock, Tom Coulthard, and colleagues). Suggest simply asserting that detachment limited is a common or popular choice."

This indeed sounds better, so **I changed the wording (line 44).**

"40 Of the three suggested reasons for the widespread use of detachment-limited discharge-slope models, I think the second two are really the important ones. The first might be a bit misleading to readers, because any of the three flavors of model discussed in this paper can be related to a power-law slope-area relationship. As far as I know, the link between erosion/transport and slope-area was actually first identified in a transport-limited context. If I recall right, Howard (1980 in Thresholds in Geomorphology) articulated a slope-area relation based on a variety of different transport formulas, and Howard and Kerby (1983) followed up with a field-based study. Then Willgoose et al. (1991) and Willgoose (1994) really hit home the slope-area relation in a transport-limited context. So having a link with Flints law is not unique to the detachment-limited formulation. The solution I suggest is just to add a sentence, maybe after the sentence following eq (6), to the effect that transport-limited and other types of erosion law can also be linked to Flints law (references), but the relationship is especially simple for the area-slope erosion law in eq (6)."

This is basically true, but nevertheless the transport-limited approach seems not be used in such a simple form as the detachment-limited approach nowadays. **I adjusted the wording so that it no longer implies that the relationship to Hack's findings is clearer, but only simpler for the transport-limited model (lines 49–50) and changed in the abstract accordingly (lines 5–6). In addition, I mentioned that Willgoose (1991) already brought the transport-limited model into the context of Hack's findings (lines 76–81)**.

"49 has become some kind of paradigm – I think I understand what you mean here, but as written it is a vague statement (what exactly constitutes a paradigm? what kind of paradigm?). Better I think to leave this comment out."

**I replaced it by "a common choice" (line 59).**

"56 little is known – this statement is a bit unfair to researchers who have tried to pin it down. Suggest softening to something like the effective value of n is less well known. You could also add something like: some studies suggest a linear scaling (REFS), some sub-linear (REFS), and some super-linear (REFS)."

I know about some fields where several highly reputed scientists spent much work, and there is still very limited knowledge. Estimating the value of $n$ is just extremely difficult and susceptible to systematic errors. Personally, I do not trust in any of the estimates of $n$ from the literature very much, but discussing this would be a different paper. **I used a softer wording now (lines 66–67) and referred to Lague (2014) for an overview (lines 70–71). In addition, I discussed one of the potential sources of systematic errors at the end of Sect. 4 now (lines 318–344).**

"62 There are quite a few other papers that report estimates of K values, which could be cited here. I guess the e.g. is meant to say there are more papers than I feel like bothering to list here, but if you want a starting point, try these two. I guess thats ok, but you are likely to annoy the authors of the ones you left out. An alternative would be to find a recent paper or two that reports K values and is reasonably comprehensive in its referencing, and cite as So-and-so, 20xx, and references therein."

The list was even not thought to be a starting point for studies about $K$ in general. The point is that the term "erodibility" could be misinterpreted as a property of the rock alone, while it is a lumped parameter. These two references particularly refer to the dependence on climate, and the "e.g." is due to that fact that I am not sure whether there are also older references addressing the dependence on climate.

"76 The wording is a bit awkward here; suggest leaving out despite increasing computing... (we all know computers have gotten faster)."

True, **I removed this phrase (line 94).**

"87 models treat "

Thanks! **Fixed (line 105).**

"100 confusing because you would choose either (2) or (8); how about (1) and either (2) or (8) "

No, we would indeed need the 3 equations in order to obtain a system of 2 partial differential equations for the variables $H$ and $q$ in this case. However, as this part is indeed a bit complicated, **I moved it to the new Sect. 5 and explained it in more detail there (lines 357–359).**

"109 the upstream"

"eq (11) and preceding text: the way this is written seems to suggest that approaching the problem from the question of how much sediment would you get from eq (6) is a requisite for deriving the method that follows. Actually, there are at least two other pathways that I can think of. I think you are more likely to sell the approach more effectively if you point out that there are several lines of evidence to support the hypothesis that the long-term sediment flux should depend on slope and drainage area. You have ar- ticulated one of them, but it seems to me it is subject to the criticism that you are using a detachment-limited concept (eq 6) to derive a transport-limited model. An alternative would be to state that previous studies have shown that sediment-transport formulas can be cast in the form of an area-slope power expression, and cite some references. You could also lean on Davy and Lague (2009) here, because when you combine their expression with a unit-stream-power detachment rate and $Q_w \propto A$ ($Q_w$ being water discharge), you end up with a transport capacity (if I recall right) that looks like $A^{\frac{3}{2}}S$ (more generally, $A^{m+1}S^n$). I think it is fair to say that there is uncertainty in the literature over how best to express transport capacity in models of stream profile evolution or landscape evolution. Some have $Q \propto Q_w S$, some (e.g., Willgoose, Howard, based on the empirical Einstein-Brown expression) have $Q \propto q_w^2 S^2$, and some include a transport threshold. Key point for your purposes is that $Q = KA^{m+1}S$ falls within the span of proposed laws."

"130 change which to that (introduces a restrictive clause)"

**Fixed (line 128).**

It indeed reads a bit as if this section started from the detachment-limited model and measured the resulting sediment yield. However, I rather thought of the generic model based on Hack's findings, i.e., on a uniform effective erosion rate that yields river profiles with constant concavity and steepness indices. At this level, it just describes an erosion rate without regard to any mechanism. **I tried to point this out more clearly in lines 141–149, mentioned the alternative approach using more physical principles in the introduction (lines 76–81), and return to this point in lines 168–171.** However, I still prefer the empirical starting point here in order to point out that both interpretations of Hack's findings are somehow straightforward.

Thanks! **Fixed (line 158).**

"132-3 I do not understand this comment about Voronoi polygons. Normally in a finite-volume solution, you would integrate flux density over the width of a cell face, whether it is a square or a Voronoi polygon or some other shape. Using $Q$ instead of $q$, with an implied sub-grid-scale channel width (I suppose), you do not need to do this integration; but that is true regardless of the shape of your cells. Is your point that the discrete representation in eq 12 works in principle for any grid mesh, regular or irregular? Consider removing this statement, as it seems like a bit of a distraction."

You are right, the Voronoi condition is not essential here, although of advantage for the accuracy. **I formulated it for a general finite-volume discretization now (lines 160–163).**

"137-8 See comment above about transport laws. Equations like (11) have been frequently used in the literature. In particular, in the work of Willgoose et al. (1991a,b,c) and subsequently, the slope-area relationship is used to estimate parameters for a transport law. If there is something very specific about eq (11) that you think is unique, then that should be pointed out. Otherwise, the statement carries the implication that transport laws have never before been derived from slope-area analysis, which is not correct."

**I clarified this point (lines 141–149 and 168–172, see also the above comment).** But beyond this, I think the simple formulation given in Eq. (11) is new, and this its advantage is explained in lines 173–180.

"144-5 I believe it is more than a matter of terminology. It is rather a matter of dimensionality. If you suppose $m = 1/2$, then the erodibility has dimensions of inverse time, whereas the transport coefficient has dimensions of length$^2$/time."

No, both have the dimension of inverse time and even the same value under identical conditions (same river profile at the same erosion rate).

"153-157 Consider adding some more explanatory text here. At first glance, eq (15) looks like a Taylor expansion to first order. But if I am following this correctly, actually $Q_{0i}$ includes the value of $Q_i$ at $t_0$ plus the partial derivative of $Q_i$ with respect to $H_i$ times the change in $H_i$ during one time step. That's a clever idea, and is consistent with the explanation on line 156, but it took me some time to work it out. Other readers might similarly misinterpret $Q_{0i}$ on a first look, and yet its definition is really key to whole scheme. Suggest devoting a full sentence or so to pointing out the definition and importance of it.

The linear dependence of all properties (height, flux) on the base level, i.e., that Eq. (15) is exact, is indeed the key to understanding the numerical scheme. **I added a detailed explanation including the new Fig. 1 where it can be recognized that it is not just a first-order Taylor approximation, but an exact relation (lines 201–207).**

"177 The challenge for readers is that the donor information is buried in the definitions of alpha and beta. Suggest adding, after the word donors, (because $\alpha$ and $\beta$ depend on donors $Q$ and $Q'$, respectively)."

Ok, seems to be helpful, **so I added something like this (lines 234–235)**.

"187, 189 - reference to a recursive implementation is vague. Suggest referring to a published algorithm(s) for sorting by downstream order."

To my knowledge there is no sorting algorithm that achieves linear complexity ($O(N)$) on average, so sorting the nodes would formally destroy the linear complexity for both transport-limited and detachment-limited erosion. Provided that a programming language that supports recursion is used, a recursive scheme is more efficient than sorting. **I briefly explained how the recursive schemes have to be designed here (lines 245–249 and 250–251)**.

"198-200 Please document somehow the specifications for the performance tests: for example, the number of iterations were run for each case.

Seriously? The values are, of course, obtained from several runs (here with 100 and 1000 time steps) and checked for consistency. However, we know that the uncertainty is primarily not due to the process of measuring, but to the details of the implementation of the individual functions. So **I added one short remark (lines 259–260)**.

"207-8 With all due respect, I think this is a missed opportunity. As noted above, I suggest trying a solution with one row of grid nodes (so, strictly one dimensional) and a uniform drainage area. Then it reduces to a linear diffusion equation, which you could compare with the transient analytical solution for diffusion given a step change at one boundary."

I am quite sure that this opportunity does not run away so soon, and there is already some ongoing work in this direction. However, the 1-D version is not interesting enough and thus not really an option.

"213 Please explain the rationale for increasing $\delta t$ over time."

Ok, **I explained the scheme a bit more in detail, although this required some reordering (lines 279–292)**.

"217-8 To avoid potential confusion, it would be useful to clarify that the two models are NOT equivalent, but rather their steady state solutions have the same slope-area relationship. Either give the predicted slope-area equivalence, or quote a reference that does (or both)."

Good point! **I discussed the condition under which the models are equivalent and how they differ at the end of Sect. 2 now (lines 178–186)**.

"223 and following: The tent-shaped uplift pattern is a clever test. I think the example would be easier to follow if you did two things. First, before referring to the results (figure 2), explain why you are using this tent-shaped uplift and what differences you expect to see between the two models. That way, the reader knows what to look for in Figures 2 and 3. Second, it would be very helpful to provide an analytical solution for the two models. You could simply use Hacks law to relate drainage area to distance (I would just make the exponent 2 for simplicity). Plot the predicted longitudinal stream profiles with a tent-shaped uplift pattern for each model, in chi space (you could do linear space too). That way there is a clear expectation for Figure 3 (actually, you could simply add the analytical profiles to Figure 3). If I have done the math right, the two profiles should be defined by

$$dH/dx = (u0(Lx)/K)x^{-hm}$$
$$dH/dx = (u0/K)x^{-h(m+1)} \int_0^x (L-x)dx$$

where $u_0$ is the uplift rate at the ridgeline, $h$ is the Hack exponent, and $L$ is the domain half-width. So, should be possible to plot these as analytical expectations."

A good idea in principle, but it runs into problems with the contribution of tributaries for the transport-limited model. In contrast to the catchment sizes, the sediment fluxes do not obey Hack's law. As an example, all single-pixel catchments are located at the ridge and thus exposed to the maximum uplift in the 1D formulation using Hack's law. In the network, however, they are distributed over the entire catchment. So they are exposed to lower uplift on average and thus provide less sediment. I recently worked on a similar problem in the context of glacial erosion (Prasicek et al., EPSL, 2020, 0.1016/j.epsl.2020.116350), and it is more complicated that it seems first.

"251 $\phi$ and $\psi$ seem to be parameters rather than functions."

This depends on the point of view. If we consider the catchment size $A$ as a variable, $\phi$ and $\psi$ are functions. **I clarified this by writing $\phi(A)$ and $\psi(A)$ at several locations where it makes the terms not too cumbersome**.

"254 Davy and Lague deserve much credit for introducing this formulation in the landscape evolution context, and showing that it relates to the earlier Beaumont model except that the length scale varies with unit discharge. For the record, similar formulations with erosion/entrainment and deposition terms seem to be widely used in the sedimentation engineering and soil erosion communities."

Good point, but I must admit that I did not work on these fields since the end of the 1990s, and I am not familiar with the recent literature.

"eq (34) I like this alternative expression of phi. Presumably it would simplify calibration by removing a built-in correlation between the two parameters."

I even used the model like this for several years in such way that $K$ defines the steepness of equilibrium profiles and $G$ let us more between detachment-limited and transport-limited. The Associate Editor also suggested to discuss this class of models in more detail, and **new part additionally contains one more alternative formulation with a different set of parameters (lines 396–419).**

"284-5 Would not $G \to \infty$ lead to $Q_i \to 0$ by eq (31)?"

Indeed! This is another argument why the Davy-Lague model must be rescaled in order to have a well-defined transition to the transport-limited end member. **This is discussed more thoroughly now in Sect. 5 (lines 365–368).**

"286 missing to "

**Fixed (line 458).**

"287 extra of "

**Fixed (line 459).**

"294 some kind of is a bit vague. Suggest re-wording to be more precise."

**This sentence has vanished by some restructuring the sections anyway (line 475).**

"295-6 Can you articulate what process(es) this kind of formulation is meant to represent? Is the idea that some of the material is so fine-grained that it will not end up being deposited until it reaches the ocean or some kind of closed basin? I wonder whether an alternative would be to build this into $dQ/dH$."

This was exactly the idea behind it. **I added a note on the idea (lines 478–479).**

"303-4 This statement is not clear to me. From the references cited, I guess that by scaling problem you mean the classic problem of grid-size scaling. Yet that wasn't mentioned as an issue with the prior models (transport limited and linear decline), so why is it more of an issue with equation 35 than with, say, equation 26?"

Admittedly, the explanation is very short. The reason why it may occur here, but not in the previous consideration is breaking the sediment balance (direct excavation) along linear river segments (line 487). It may become clear to the readers if they proceed to the reference Hergarten (2020a). However, it is just thought as a warning for those who ever get to the point where they want to implement such a model.

"317-325 Are you suggesting to solve for diffusive flux in the flow directions using the implicit scheme, and the other directions using some other scheme? How would you avoid double-counting the fluxes in the cardinal flow directions? Overall, I think the sketch presented here for handling diffusion is not really convincing. I would recommend either deleting it, or expanding it to really demonstrate how it would work."

Exactly! I thought it would be not a big challenge to follow the idea, but **I explained it in more detail now (lines 504–519)**.

**Associate Editor (Jean Braun)**

"I think the author could provide a more quantitative comparison between the results obtained with this new method and those obtained by solving the Davy and Lague (2009) approach directly (i.e. without using the flux divergence formulation) as done by Yuan et al (2019). I have rapidly coded the two approaches in 1D and easily showed that the improvement in speed is a strong function of $G$ (for $G \geq 1$).

You are right that the improvement in performance compared to the iterative implementation of Yuan et al. (2019) rapidly increases with increasing $G$ for $G > 1$ as the convergence of the iterative scheme slows down. The reason why I did not consider this in such detail is that the estimates of Yuan et al. (2019) might even be too pessimistic. I used basically the same approach for some years, but with a fixed number of 2 iterations (the minimum number of flow directions change), and did not care for convergence. I also work reasonably well, in particular if we take into account that changes in flow directions require a limitation of $\delta t$ anyway. That was the reason why I stayed on a rather qualitative level. i.e., focusing on the advantage that we do not have to take care of anything when using the direct scheme. Nevertheless, **I introduced some numbers on the gain obtained from comparing the results of Yuan et al. (2019) and Guerit et al. (2019)**.

"Additionally to the suggestions made by the reviewers, I would like the author to provide a more structured explanation of the procedure used in the solution of what the author calls the 'linear decline model'. I note that the author's implementation is based on equations [1]+[26] (I use square brackets to indicate equation numbers from the manuscript and parentheses to indicate equation numbers from this comment):

$$\frac{\partial H}{\partial t} = U - \phi S + \psi Q \quad (1)$$

and its discretised form [28]; but, of course, it also uses equation [17], which is equivalent to [16], itself a discretised form of equation:

$$\frac{\partial H}{\partial t} = U \mathrm{div} q \quad (2)$$

This leads me to conclude that the author uses two evolution equations simultaneously at every point of the landscape. Combining them also suggests that:

$$\mathrm{div} q = \phi S - \psi Q \quad (3)$$

This may explain why the method is hybrid, depending on the value of the $\psi$ parameter (purely advective if $\psi = 0$ and divergence based (or diffusive) for large values of $\psi$), as pointed out by the author. However, it is not clear to me how to connect all these points together. I would appreciate if the authors could help me (and other readers) clarify this point with a more structured presentation of the basic partial differential equation(s?) that are solved in the linear decline model algorithm.

Exactly! **As the main change to the manuscript, I devoted an own section to the linear decline model now (new Sect. 5). This section starts from the theoretical point of view – two coupled partial differential equations and their "usual" formulation as a single integro-differential equation. Then it addresses the limiting cases where it turns into a single differential equation, however, of different types for the detachment-limited and transport-limited end members. As a little extension, I introduced a formulation in terms of two different parameters $K_d$ and $K_t$, which can be interpreted as shared stream power.** I hope you find this section useful.

In addition, I extended the results section by Fig. 5 (lines 318–344) and some text. I did this mainly because Reviewer 3 brought the exponent $n$ into play. In the recent literature, there seems to be some trend towards rather high values of $n$. Taking into account the results from the simple tent-shaped uplift pattern for transport-limited erosion, I am wary about these large values of $n$, so I discussed the results of this example in this context as some kind of warning. Maybe you could take a look at the reasoning and let me know whether you find it useful. It is not an essential part and could easily be skipped.

Best regards,

Stefan

Stefan Hergarten

[revised manuscript text omitted]

---

## Author Response (AR2)

Dear Jean,

ok, yesterday I performed some computations and analyses on the numerical accuracy. I guess you have noticed that I really do not like the 1D version with constant diffusivity because the central point is the dendritic structure where the diffusivity strongly increases in downstream direction.

I have now analyzed the numerical error for the example with the $5000 \times 5000$ grid and introduced an additional section about the accuracy. In turn, I moved the discussion about the changes in flow directions to the new section. I hope it is helpful for the readers to see that the numerical accuracy of the scheme itself is indeed no problem at all, and caution is only required concerning changes in flow directions.

Best regards,

[revised manuscript text omitted]